# Improved Differentially Private Riemannian Optimization: Fast Sampling and Variance Reduction

**Saiteja Utpala**                                                    *saitejautpala@gmail.com*
*Independent*

**Andi Han**                                                          *andi.han@sydney.edu.au*
*University of Sydney*

**Pratik Jawanpuria**                                      *pratik.jawanpuria@microsoft.com*
*Microsoft India*

**Bamdev Mishra**                                              *bamdevm@microsoft.com*
*Microsoft India*

**Reviewed on OpenReview:** *https://openreview.net/forum?id=paguBNtqiO*

## Abstract

A common step in differentially private (DP) Riemannian optimization is sampling from the (tangent) Gaussian distribution as noise needs to be generated in the tangent space to perturb the gradient. In this regard, existing works either use the Markov chain Monte Carlo (MCMC) sampling or explicit basis construction based sampling methods on the tangent space. This becomes a computational bottleneck in the practical use of DP Riemannian optimization, especially when performing stochastic optimization. In this paper, we discuss different sampling strategies and develop efficient sampling procedures by exploiting linear isometry between tangent spaces and show them to be orders of magnitude faster than both the MCMC and sampling using explicit basis construction. Furthermore, we develop the DP Riemannian stochastic variance reduced gradient algorithm and compare it with DP Riemannian gradient descent and stochastic gradient descent algorithms on various problems.

## 1 Introduction

Differential privacy (DP) provides a rigorous treatment for the notion of data privacy by precisely quantifying the deviation in the model's output distribution under modification of a small number of data points (Dwork et al., 2006b). Provable guarantees of DP coupled with properties like immunity to arbitrary post-processing, and graceful composability have made it a de-facto standard of privacy with steadfast adoption in the real world (Erlingsson et al., 2014; Apple, 2017; Near, 2018; Abowd, 2018). Furthermore, it has been shown empirically that DP models resist various kinds of leakage attacks that may cause privacy violations (Rahman et al., 2018; Carlini et al., 2019; Sablayrolles et al., 2019; Zhu et al., 2019; Balle et al., 2022; Carlini et al., 2022).

Various approaches have been explored in the literature to ensure differential privacy in machine learning models. These include output perturbation (Chaudhuri et al., 2011; Zhang et al., 2017) and objective perturbation (Chaudhuri et al., 2011; Iyengar et al., 2019), in which a perturbation term is added to the output of a non-DP algorithm or the optimization objective, respectively. Another approach, gradient perturbation, involves perturbing the gradient information at every iteration of gradient based approaches and has received significant interest in the context of deep learning and stochastic optimization (Song et al., 2013; Bassily et al., 2014; Abadi et al., 2016; Wang et al., 2017; Bassily et al., 2019; Wang et al., 2019a; Bassily et al., 2021).

Recently, achieving differential privacy over Riemannian manifolds has also been explored in the context of obtaining Fréchet mean (Reimherr et al., 2021; Utpala et al., 2022b; Soto et al., 2022) and, more generally, solving empirical risk minimization problems (Han et al., 2022a). Riemannian geometry is a generalization of the Euclidean geometry (Lee, 2006) and includes several non-linear spaces such as the set of positive definite matrices (Bhatia, 2009), set of orthogonal matrices (Edelman et al., 1998), and hyperbolic space (Beltrami, 1868; Gromov, 1987), among others. Many machine learning tasks such as principal component analysis (Absil et al., 2007), low-rank matrix/tensor modeling (Boumal & Absil, 2011; Kasai et al., 2019; Jawanpuria & Mishra, 2018; Nimishakavi et al., 2018), metric learning (Bhutani et al., 2018; Han et al., 2021), natural language processing (Jawanpuria et al., 2019a; 2020a;b), learning embeddings (Nickel & Kiela, 2017; 2018; Suzuki et al., 2019; Jawanpuria et al., 2019b; Qi et al., 2021), optimal transport (Mishra et al., 2021; Jawanpuria et al., 2021; Han et al., 2022b), etc., may be viewed as problem instances on Riemannian manifolds.

In differentially private Riemannian optimization (Han et al., 2022a), a key step is to use tangent Gaussian sampling at every iteration to perturb the gradient direction in the tangent space. Han et al. (2022a) proposed to use the Markov Chain Monte Carlo (MCMC) method (Robert & Casella, 1999), which is computationally expensive especially on matrix manifolds with large dimensions. When the underlying Riemannian metric is induced from the Euclidean metric, such as for sphere, Han et al. (2022a) showed one can avoid MCMC via basis construction for the tangent space. For general manifolds of interest, however, a discussion on basis construction and computationally efficient sampling is missing. The sampling step is computationally prohibitive, especially when performing differentially private stochastic optimization over Riemannian manifolds, where the number of sampling calls is relatively high compared to the case of deterministic optimization. It should also be noted that generalizing more sophisticated differentially private Euclidean stochastic algorithms like differentially private stochastic variance reduced gradient (Wang et al., 2017) to Riemannian geometry is non-trivial and is an active area of research. The benefits of (non-private) Riemannian stochastic variance reduction gradient (RSVRG) methods over Riemannian stochastic gradient (Bonnabel, 2013) has been studied in existing works (Zhang et al., 2016; Zhou et al., 2019; Han & Gao, 2021; Sato et al., 2019).

Our main contributions on improving differentially private Riemannian optimization framework are summarized below.

1. **Sampling.** We propose generic fast sampling methods on the tangent space for various matrix manifolds of interest. This makes differentially private Riemannian optimization more practically appealing for real-world applications. The proposed sampling strategy is based on linear isometry between tangent spaces. We show that it is computationally efficient and orders of magnitude faster than other sampling schemes, MCMC, and explicit basis construction, presented in (Han et al., 2022a).

2. **DP-SVRG.** We propose a differentially private Riemannian stochastic variance reduced gradient (DP-RSVRG), enriching the suite of differentially private stochastic Riemannian optimization methods.

Our first contribution allows to scale differentially private Riemannian optimization algorithms since sampling is now faster. Our second contribution is on developing DP-RSVRG and together with faster sampling, DP-RSVRG is scalable to large datasets. In the experiments section, we show the empirical benefit of both of these contributions.

**Organization.** The rest of the paper is organized as follows. Section 2 gives a background on Riemannian geometry, Riemannian optimization, and differential privacy. We then use various properties of tangent Gaussian distribution and discuss different possible sampling strategies in Section 3. Section 4 presents our proposed sampling procedure and gives exact details about how to implement it in practice for several manifolds of interest. In Section 5, we develop a differentially private Riemannian stochastic variance reduction gradient algorithm (DP-RSVRG). Section 6 discusses the empirical results. Section 7 concludes the paper.

## 2 Preliminaries and related work

**Riemannian Geometry.** A Riemannian Manifold $\mathcal{M}$ of dimension $d$ is smooth manifold with an inner product structure $\langle .,. \rangle_w$ (i.e., having a Riemannian metric) on every tangent space $T_w\mathcal{M}$. Given a basis $\mathscr{B} = (\beta_1, \ldots, \beta_d)$ for $T_w\mathcal{M}$ at $w \in \mathcal{M}$, the Riemannian metric can be represented as a symmetric positive definite matrix $G_w$ and the inner product can be written as

$$\langle \nu_1, \nu_2 \rangle_w = \overrightarrow{\nu_1}^T G_w \overrightarrow{\nu_2},$$

where $\overrightarrow{\nu_1}, \overrightarrow{\nu_2}$ are coordinates of the tangent vectors $\nu_1, \nu_2 \in T_w\mathcal{M}$ in the coordinate system given by $\mathscr{B}$. An induced norm is defined as $\|\nu\|_w = \sqrt{\langle \nu, \nu \rangle_w}$ for $\nu \in T_w\mathcal{M}$. Let $\gamma : [0,1] \to \mathcal{M}$ denote any smooth curve and $\gamma'(t) \in T_{\gamma(t)}\mathcal{M}$ its derivative, then distance between $w_1, w_2 \in \mathcal{M}$ is defined as $\mathrm{dist}(w_1, w_2) = \inf_{\gamma : \gamma(0) = w_1, \gamma(1) = w_2} \int_0^1 \|\gamma'(t)\|_{\gamma(t)} \, dt$.

A smooth curve $\gamma : [0,1] \to \mathcal{M}$ is called the geodesic if it locally minimizes the distance between $\gamma(0)$ and $\gamma(t)$. For any $\nu \in T_w\mathcal{M}$, the exponential map is defined as $\mathrm{Exp}_w(\nu) = \gamma(1)$ where $\gamma(0) = w$ and $\gamma'(0) = \nu$. If between any two points $w, w' \in \mathcal{W} \subseteq \mathcal{M}$ there is a unique geodesic connecting them, then the exponential map has an inverse map $\mathrm{Exp}_w^{-1} : \mathcal{W} \to T_w\mathcal{M}$, which maps a point on the manifold to the tangent space $T_w\mathcal{M}$. Transporting the vectors on the manifold requires the notion of parallel transport. In particular, the parallel transport from $w_1 \in \mathcal{M}$ to $w_2 \in \mathcal{M}$ denoted as $\mathrm{PT}^{w_1 \to w_2} : T_{w_1}\mathcal{M} \to T_{w_2}\mathcal{M}$ is a linear isometry (i.e., inner product preserving) along a geodesic. In this work, the curvature of a manifold refers to the sectional curvature, which provides a local measure of curvature at each point on the manifold.

The Riemannian gradient of a real valued function $f : \mathcal{M} \to \mathbb{R}$, denoted as $\mathrm{grad}\, f(w)$, is a tangent vector such that for any $\nu \in T_w\mathcal{M}$,

$$\langle \mathrm{grad}\, f(w), \nu \rangle_w = \mathrm{D}f[w](\nu),$$

where $\mathrm{D}f[w](\nu)$ denotes the directional derivative of $f$ at $w$ along $\nu$. We refer the readers to (Do Carmo & Flaherty Francis, 1992; Lee, 2006) for a detailed exposition of Riemannian geometry and (Absil et al., 2009; Boumal, 2022) for Riemannian optimization.

**Function classes on Riemannian Manifolds.** We call a neighbourhood $\mathcal{W} \subseteq \mathcal{M}$ totally normal if for any two points, the exponential map is invertible. Let $\mathcal{W} \subseteq \mathcal{M}$ be a totally normal neighborhood and $D_{\mathcal{W}}$ be its diameter and $\kappa_{\min}$ be the lower bound on the sectional curvature of $\mathcal{W}$.

A function $f : \mathcal{W} \to \mathbb{R}$ is called $L_0$-geodesically Lipschitz continuous (L-g-lipschitz) if for any $w_1, w_2 \in \mathcal{M}$, $|f(w_1) - f(w_2)| \leq L_0 \mathrm{dist}(w_1, w_2)$. Under the assumption of continuous gradient, a function $f$ is $L_0$ geodesically Lipschitz continuous if and only if

$$\|\mathrm{grad}\, f(w)\| \leq L_0,$$

for all $w \in \mathcal{M}$ (Boumal, 2022). A differentiable function $f : \mathcal{M} \to \mathbb{R}$ is geodesically L-smooth (L-g-smooth) if its gradient is $L$-Lipschitz, i.e., $\|\mathrm{grad}\, f(w_1) - \mathrm{PT}^{w_2 \to w_1} \mathrm{grad}\, f(w_2)\|_{w_1} \leq L \mathrm{dist}(w_1, w_2)$. Additionally, it can be shown that if $f$ is geodesically L-smooth, then

$$f(w_1) \leq f(w_2) + \langle \mathrm{grad}\, f(w_2), \mathrm{Exp}_{w_2}^{-1}(w_1) \rangle_{w_2} + \frac{L}{2} \|\mathrm{Exp}_{w_2}^{-1}(w_1)\|_{w_2}^2,$$

for all $w_1, w_2 \in \mathcal{M}$. A function $f$ is called geodesically $\mu$-strongly convex ($\mu$-strongly g-convex) (Zhang et al., 2016) if for all $w_1, w_2 \in \mathcal{W}$, it satisfies

$$f(w_1) \geq f(w_2) + \langle \mathrm{grad}\, f(w_2), \mathrm{Exp}_{w_2}^{-1}(w_1) \rangle_{w_2} + \frac{\mu}{2} \|\mathrm{Exp}_{w_2}^{-1}(w_1)\|_{w_2}^2.$$

Let $w^*$ be a global minimizer of $f$. Then $f : \mathcal{W} \to \mathbb{R}$ is said to satisfy the Riemannian Polyak–Łojasiewicz (PL) condition if there exists $\tau > 0$, such that,

$$f(w) - f(w^*) \leq \tau \|\mathrm{grad}\, f(w)\|_w^2,$$

for any $w \in \mathcal{M}$(Zhang et al., 2016). The Riemannian PL condition is a strictly weaker notion than the geodesic strong convexity, i.e., every geodesic $\mu$-strongly convex function satisfies the Riemannian PL condition (with $\tau = 1/(2\mu)$) and there exist functions that satisfy the Riemannian PL condition but are not geodesically strongly convex. The trigonometric distance bound from Zhang & Sra (2016) (see Lemma 10), which is crucial for deriving convergence analysis of Riemannian optimization algorithms makes use of the curvature constant, is defined as

$$\zeta = \begin{cases} \frac{\sqrt{|\kappa_{\min}|}D_{\mathcal{W}}}{\tanh\left(\sqrt{|\kappa_{\min}|}D_{\mathcal{W}}\right)} & \kappa_{\min} < 0 \\ 1 & \kappa_{\min} \geq 0. \end{cases}$$

**Differential privacy.** Let $\mathcal{Z}$ be an input data space and two datasets of size $Z, Z' \in \mathcal{Z}^n$ of size $n$ are called *adjacent* if they differ by at most one element. We represent the adjacent datasets $Z, Z'$ by the notation $Z \sim Z'$. A manifold-valued randomized mechanism $\mathcal{R} : \mathcal{Z}^n \to \mathcal{M}$ is said to be $(\epsilon, \delta)$-approximately differentially private (ADP) (Dwork et al., 2006a; Wasserman & Zhou, 2010) if for any two adjacent datasets $Z \sim Z'$ and for all measurable sets $S \subseteq \mathcal{M}$, we have

$$\mathbb{P}\left[\mathcal{R}(Z) \in S\right] \leq \exp\left(\epsilon\right)\mathbb{P}\left[\mathcal{R}(Z') \in S\right] + \delta.$$

*Rényi differential privacy* (RDP) by Mironov (2017) is a refinement of DP which gives tight privacy bounds under composition of mechanisms. The $\lambda$-*th moment* of a mechanism $\mathcal{R}$ is defined as

$$\mathcal{K}_{\mathcal{R}}(\lambda) = \sup_{Z \sim Z'} \log\left(\mathop{\mathbb{E}}_{o \sim \mathcal{R}(Z)}\left[\left(\frac{p(\mathcal{R}(Z) = o)}{p(\mathcal{R}(Z') = o)}\right)^{\lambda}\right]\right),$$

and mechanism $\mathcal{R}$ is said to satisfy $(\lambda, \rho)$-RDP if $\frac{1}{\lambda-1}\mathcal{K}_{\mathcal{R}}(\lambda - 1) \leq \rho$. If the mechanism $\mathcal{R} : \mathcal{Z} \to \mathcal{M}$ is an (adaptive) composition of $k$ mechanisms $\{\mathcal{R}_i\}_{i=1}^{k}$, i.e., $\mathcal{R}_i : \prod_{j=1}^{i-1} \mathcal{M}_j \times \mathcal{Z}^n \to \mathcal{M}_i$, then

$$\mathcal{K}_{\mathcal{R}}(\lambda) \leq \sum_{i=1}^{k} \mathcal{K}_{\mathcal{R}_i}(\lambda).$$

Using the moments accountant technique (Abadi et al., 2016), $(\lambda, \rho)$-RDP mechanism can be given $(\epsilon, \delta)$-ADP certificate. We refer the interested readers to (Dwork et al., 2014; Vadhan, 2017) for more details.

**Differential privacy on Riemannian manifolds.** Reimherr et al. (2021) are the first to consider differential privacy in the Riemannian setting and derived the Riemannian Laplace mechanism based on distribution from (Hajri et al., 2016). Utpala et al. (2022b) derive output perturbation for manifold of symmetric positive definite matrices (SPD) with the Log-Euclidean metric based on distribution from (Schwartzman, 2016). While (Reimherr et al., 2021; Utpala et al., 2022b) focus on output perturbation, Han et al. (2022a) propose a unified differentially private Riemannian optimization framework through gradient perturbation.

Han et al. (2022a) consider the following problem (1) where the parameter of interest lies on a Riemannian manifold $\mathcal{M}$ and $\{z_i\}, i = 1, \dots, n$, represent the set of data samples, i.e.,

$$\min_{w \in \mathcal{M}}\left\{F(w) = \frac{1}{n}\sum_{i=1}^{n} f_i(w) = \frac{1}{n}\sum_{i=1}^{n} f(w; z_i)\right\}. \tag{1}$$

The aim of differentially private Riemannian optimization is to privatize the solution from a Riemannian optimization solver by injecting noise to the Riemannian gradient similar to the Euclidean case. The Riemannian gradient grad $F(w)$ belongs to the tangent space $(T_w\mathcal{M}, \langle, \rangle_w)$ and to perturb the Riemannian gradient, Han et al. (2022a) define an intrinsic Gaussian distribution on the tangent space $T_w\mathcal{M}$ with density

$$p(\nu) \propto \exp(-\|\nu - \mu\|_w^2 / 2\sigma^2), \quad \nu \in T_w\mathcal{M},$$

and refer to it as the tangent Gaussian distribution. They propose differentially private Riemannian gradient and Riemannian stochastic gradient descent algorithms.

---

**Algorithm 1:** Sampling from tangent Gaussian: a general algorithm

   **Input**  : Manifold $\mathcal{M}$ of dimension $d$, base point $w \in \mathcal{M}$, Riemannian metric $\langle .,. \rangle_w$, mean $\mu \in \mathcal{M}$ and standard deviation $\sigma > 0$.

   **Output:** $\xi$ such that $\xi \sim \mathcal{N}_w(0, \sigma^2)$.

**1** Construct a basis $\mathscr{B}$ of $T_w\mathcal{M}$ orthonormal wrt to $\langle .,. \rangle_w$.

**2** Generate $d$-dimensional coordinates $\mathbf{a} \sim \mathcal{N}(0, \sigma^2 \mathbf{I}_d)$.

**3** Generate the sample $\xi \in T_w\mathcal{M}$ as $\xi = \sum_{i=1}^{d} a_i \beta_i$.

---

## 3 Sampling from tangent Gaussian

In this section, we derive various properties of the tangent Gaussian distribution to discuss different sampling strategies for different manifolds. The proofs of the claims discussed in this section are provided in Appendix B.

We begin with the definitions of the Lebesgue measure on the tangent space and the tangent Gaussian distribution. We then show that the tangent Gaussian reduces to the multivariate Gaussian when an appropriate basis is constructed. This allows sampling to be performed in the intrinsic coordinates of the tangent space and then generate a tangent Gaussian sample by a linear combination of the basis vectors with the sampled coordinates.

**Definition 1** (Lebesgue measure on tangent space). *Consider a Riemannian manifold $\mathcal{M}$ with the intrinsic dimension $d$. For $w \in \mathcal{M}$, let $\mathscr{B} = \{\beta_1, \ldots, \beta_d\}$ be an orthonormal basis of $T_w\mathcal{M}$ with respect to the Riemannian metric $\langle .,. \rangle_w$. Define $\phi_{\mathscr{B}} : \mathbb{R}^d \to T_p\mathcal{M}$ as $\phi_{\mathscr{B}}(c_1, \ldots, c_d) = \sum_{i=1}^{d} c_i \beta_i$. Let $\lambda$ denote the standard Lebesgue measure on $\mathbb{R}^d$. Then, we define the Lebesgue measure on $T_w\mathcal{M}$ as the pushforward measure $\phi_*^{\mathscr{B}}$ given by*

$$(\phi_*^{\mathscr{B}} \lambda)(S) \triangleq \lambda\left(\phi_{\mathscr{B}}^{-1}(S)\right).$$

*Remark* 1. Let $\mathscr{B}_1, \mathscr{B}_2$ be two orthonormal bases of $T_w\mathcal{M}$, then $\phi_*^{\mathscr{B}_1} \lambda = \phi_*^{\mathscr{B}_2} \lambda$ because the Lebesgue measure is invariant under orthogonal transformations (with respect to the Riemannian metric). Hence, in the rest of this draft, we drop the superscript $\mathscr{B}$ for clarity and denote the pushforward measure as $\phi_* \lambda$.

We now define the tangent space Gaussian distribution (Han et al., 2022a) under the measure in Definition 1.

**Definition 2** (Tangent Gaussian (Han et al., 2022a)). *Let $w \in \mathcal{M}$, a random tangent vector $\xi \in T_w\mathcal{M}$ follows a tangent space Gaussian distribution at $w$, denoted as $\xi \sim \mathcal{N}_w(\mu, \sigma^2)$ with mean $\mu \in T_w\mathcal{M}$ and standard deviation $\sigma > 0$ if its density is given by*

$$p_w(\nu) = C_{w,\sigma} \exp\left(-\frac{\|\nu - \mu\|_w^2}{2\sigma^2}\right),$$

*under the pushforward measure given in Definition 1.*

**Lemma 1.** *Let $w \in \mathcal{M}$ and $\mathscr{B}$ be any orthonormal basis of $T_w\mathcal{M}$. Also, let $\xi \in T_w\mathcal{M}$ denote a random tangent vector. Then, the following holds:*

    *1. If $\xi \sim \mathcal{N}_w(\mu, \sigma^2)$ for some $\mu \in T_w\mathcal{M}$ and for $\sigma > 0$, then $C_{w,\sigma} = (2\pi\sigma^2)^{1/2}$.*

    *2. $\xi \sim \mathcal{N}_w(\mu, \sigma^2) \iff \vec{\xi} \sim \mathcal{N}(\vec{\mu}, \sigma^2 I_d)$ where $\vec{\xi}, \vec{\mu} \in \mathbb{R}^d$ denote coordinates in basis $\mathscr{B}$.*

    *3. If $\xi \sim \mathcal{N}_w(0, \sigma^2)$, then $\mathbb{E} \|\xi\|_w^2 = d\sigma^2$, where $d$ is the dimension of the manifold.*

*Remark* 2. Statement 3 of Lemma 1 improves the bound on variance from (Han et al., 2022a, Lemma 4) by removing the dependency on the metric tensor $G_w$.

Statement 2 of Lemma 1 implies that a random tangent vector follows tangent Gaussian if and only if its random coordinates in any orthonormal basis follow from the Euclidean Gaussian distribution of the intrinsic dimension. This allows to avoid the computationally expensive MCMC based sampling, which is suggested

---

**Algorithm 2:** Sampling from tangent Gaussian using isometric transportation

> **Input** : Manifold $\mathcal{M}$ of dimension $d$, base point $w \in \mathcal{M}$, Riemannian metric $\langle .,.\rangle_p$, mean $\mu \in$ standard deviation $\sigma > 0$, reference point $\widehat{w}$.
>
> **Output:** $\xi$ such that $\xi \sim \mathcal{N}_w(0, \sigma^2)$.
>
> **1** Sample $d$ coordinates $\mathbf{a} \sim \mathcal{N}(0, \sigma^2 \mathbf{I}_d)$.
>
> **2** Generate the sample $\zeta \in T_{\widehat{w}}\mathcal{M}$ at $\widehat{w}$.
>
> **3** Generate the sample $\xi \in T_w\mathcal{M}$ by isometric transportation of $\zeta$ from $T_{\widehat{w}}\mathcal{M}$ to $T_w\mathcal{M} : \xi = \mathrm{LI}^{\widehat{w}\to w}(\zeta)$.

---

in (Han et al., 2022a) for manifolds with non-Euclidean Riemannian metrics, and instead apply the basis construction approach for any manifold. We summarize the procedure in Algorithm 1. However, the practical efficiency depends on how Steps 1, 2, and 3 of Algorithm 1 are implemented for different manifolds.

One natural approach for sampling from tangent Gaussian is to perform an explicit basis construction (Step 1) in which we fully enumerate the basis elements in $\mathscr{B}$. Steps 2 and 3 can subsequently be performed in a straightforward manner. The other approach is to combine Steps 1, 2, and 3 implicitly. We discuss these approaches in the following sections.

### 3.1 Sampling with explicit basis enumeration

Here, we construct a basis explicitly, i.e., either analytically or by using Gram-Schmidt orthogonalization.

**Gram-Schmidt orthogonalization.** The tangent space at a point on a manifold is parameterized by a system of linear equations. One approach to perform sampling is to first solve the underlying linear equations to get the basis $\mathscr{B}$ of $T_w\mathcal{M}$ that is orthonormal in the sense of the Euclidean metric. Depending on the Riemannian metric, we now have two further scenarios.

- When the Riemannian metric $\langle .,.\rangle_w$ is a scaled Euclidean metric, then the orthonormal basis with respect to the Riemannian metric $\langle .,.\rangle_w$ can be obtained by appropriate scaling of $\mathscr{B}$.

- If the Riemannian metric $\langle .,.\rangle_w$ is a more general metric, we employ the Gram-Schmidt (GS) orthogonalization process on $\mathscr{B}$ to generate a new basis that is orthonormal with respect to the Riemannian metric $\langle .,.\rangle_w$. This is computationally expensive because if $d$ is the dimension of manifold, then we have to evaluate $\mathcal{O}(d^2)$ inner products $\langle .,.\rangle_w$.

**Analytic basis construction.** One way to avoid the computationally prohibitive GS orthogonalization strategy is to analytically construct bases for different manifolds. This can be done for various manifolds by exploiting the geometry of the space. We construct the full orthonormal basis with respect to the metric $\langle .,.\rangle_w$ explicitly by full enumeration. We empirically observe (refer Section 6) that sampling with the explicit basis construction strategy is computationally expensive even if the basis is known analytically.

### 3.2 Sampling implicitly using isometric transportation

Since our end goal is to efficiently generate tangent Gaussian samples, instead of first fully constructing the orthonormal basis and then performing linear combinations, we aim to combine Steps 1, 2, and 3 of Algorithm 1. Hence, we do not fully enumerate the basis but rather create a basis implicitly.

Given a manifold with a Riemannian metric and depending on the basis chosen, there are many ways of implementing the implicit basis strategy. We propose a unified way that is both computationally efficient and easy to implement using linear isometric transportation between tangent spaces.

The key observation of this strategy is the following claim which states that to sample from the tangent Gaussian on $T_w\mathcal{M}$ for $w \in \mathcal{M}$, one can simply sample from the tangent Gaussian from any other base (reference) point $\widehat{w}$ and then transport the sample using any linear isometry operator from the reference point $\widehat{w}$ to the required base point $w$.

Table 1: Reference points $\widehat{w}$ for Algorithm 2. $\mathbf{I} \in \mathbb{R}^{m \times m}$ denotes the identity matrix. $(\mathbf{e}_1, \ldots, \mathbf{e}_r)$ denotes the standard basis vectors of $\mathbb{R}^m$ and $\mathbf{o} \in \mathbb{R}^m$ denotes the zero vector. $\langle , \rangle_F$ and $\langle , \rangle_2$ denote the standard Euclidean inner product on matrices and vectors, respectively. We observe that at specific reference points both the Riemannian metric and tangent space expressions simply.

| Manifold | Metric | Reference point $\widehat{w}$ | Tangent space $T_{\widehat{w}}\mathcal{M}$ | Metric $\langle , \rangle_{\widehat{w}}$ | Algorithm | Cost |
|---|---|---|---|---|---|---|
| SPD | Affine-Invariant | $\mathbf{I} \in \mathbb{R}^{m \times m}$ | $\mathrm{SYM}(m)$ | $\langle , \rangle_F$ | Alg 3 | $\mathcal{O}(m^3)$ |
| | Bures-Wasserstein | $\mathbf{I} \in \mathbb{R}^{m \times m}$ | $\mathrm{SYM}(m)$ | $\langle , \rangle_F/4$ | Alg 4 | $\mathcal{O}(m^3)$ |
| | Log-Euclidean | $\mathbf{I} \in \mathbb{R}^{m \times m}$ | $\mathrm{SYM}(m)$ | $\langle , \rangle_F$ | Alg 5 | $\mathcal{O}(m^3)$ |
| Hyperoblic | Poincaré ball | $\mathbf{o} \in \mathbb{R}^m$ | $\mathbb{R}^m$ | $\langle , \rangle_2$ | Alg 6 | $\mathcal{O}(m)$ |
| | Lorentz hyperboloid | $\mathbf{e}_1 \in \mathbb{R}^m$ | $\{0\} \times \mathbb{R}^{m-1}$ | $\langle , \rangle_2$ | Alg 7 | $\mathcal{O}(m)$ |
| Sphere | Euclidean | $\mathbf{e}_1 \in \mathbb{R}^m$ | $\{0\} \times \mathbb{R}^{m-1}$ | $\langle , \rangle_2$ | Alg 8 | $\mathcal{O}(m)$ |
| Stiefel | Euclidean | $[\mathbf{e}_1, \ldots, \mathbf{e}_r] \in \mathbb{R}^{m \times r}$ | $\mathrm{SKEW}(r) \times \mathbb{R}^{(m-r) \times r}$ | $\langle , \rangle_F$ | Alg 9 | $\mathcal{O}(mr^2)$ |
| Grassmann | Euclidean | $[\mathbf{e}_1, \ldots, \mathbf{e}_r] \in \mathbb{R}^{m \times r}$ | $\{0\}^{r \times r} \times \mathbb{R}^{(m-r) \times r}$ | $\langle , \rangle_F$ | Alg 10 | $\mathcal{O}(mr^2)$ |

**Claim 2.** *Let $\widehat{w} \in \mathcal{M}$ and let $\mathrm{LI}^{\widehat{w} \to w} : T_{\widehat{w}}\mathcal{M} \to T_w\mathcal{M}$ be any linear isometric transportation. If $\xi \sim \mathcal{N}_{\widehat{w}}(\mu, \sigma^2)$ for some $\mu \in T_w\mathcal{M}$ and $\sigma > 0$, then*

$$\mathrm{LI}^{\widehat{w} \to w}(\xi) \sim \mathcal{N}_w(\mathrm{LI}^{\widehat{w} \to w}(\mu), \sigma^2).$$

*Remark* 3. Linear isometry, as defined above, encompasses the parallel transport and more generally some classes of vector transport on manifolds (Absil et al., 2009; Boumal, 2022; Huang et al., 2015; 2017). We denote the parallel transport operation as PT and the vector transport operation as VT.

We choose the reference point $\widehat{w}$ such that it is relatively easy to sample from the tangent Gaussian at $T_{\widehat{w}}\mathcal{M}$, and then, isometrically transport from $\widehat{w}$ to the required point $w$. To be precise, we choose a reference point where two things happen: $(i)$ the tangent space $T_{\widehat{w}}\mathcal{M}$ is parametrized freely and $(ii)$ the underlying Riemannian metric $\langle ., . \rangle_{\widehat{w}}$ becomes a scaled Euclidean metric. We term this procedure as isometric transportation and summarize it in Algorithm 2. The isometric transportation strategy can be seen as performing implicit basis construction, i.e., transporting the samples from the $\widehat{w}$ to the required point $w$ is equivalent to transporting the tangent space basis from $\widehat{w}$ to $w$ as $\xi = \mathrm{LI}^{\widehat{w} \to w}\left(\sum_{i=1}^d a_i \beta_i\right) = \sum_{i=1}^d a_i \mathrm{LI}^{\widehat{w} \to w}(\beta_i)$.

Efficient implementations of these isometric transportation procedures (parallel transport and vector transport) are extensively studied in the literature (Absil et al., 2009; Xie et al., 2013; Huang et al., 2015; 2017; Thanwerdas & Pennec, 2023; Guigui & Pennec, 2022) and are readily available in many of the existing Riemannian optimization libraries (Boumal et al., 2014; Townsend et al., 2016; Miolane et al., 2020; Utpala et al., 2022a). Hence, a benefit of the isometric transportation strategy (Algorithm 2) is that one only needs to take care of the sampling at $\widehat{w}$ and the rest follows through. As we see later that implementing tangent Gaussian sampling at a properly chosen reference point $\widehat{w}$ can be made computationally efficient. For all the manifolds, sampling at $\widehat{w}$ amounts to simply reshaping samples from the standard normal distribution to certain a size and is readily implementable.

## 4 Isometric transportation based sampling for different manifolds

In this section, we discuss the proposed sampling strategy and provide details about how to implement it for several interesting manifolds. The rest of the section deals with how to concretely implement Algorithm 2 for several manifolds of interest. For each manifold, we include a summary of the reference points, the metric at the points, and the concrete algorithm for sampling in Table 1. For the expressions of the parallel transport and vector transport operations on different manifolds, see Appendix A. We illustrate through the experiments that Algorithm 2 is significantly better than other discussed procedures in computational efficiency and renders implementation of differentially private optimization computationally viable, especially for high dimensional matrix manifolds.

### 4.1 SPD manifold

Let $\mathrm{SPD}(m)$ denote the set of symmetric positive definite matrices of size $m \times m$. At $\mathbf{W} \in \mathrm{SPD}(m)$, the tangent space at $\mathbf{W}$ is $T_{\mathbf{W}}\mathrm{SPD}(m) = \mathrm{SYM}(m)$, where $\mathrm{SYM}(m)$ denotes the set of symmetric matrices of size $m \times m$. (Bhatia, 2009). We consider three Riemannian metrices: the Affine-Invariant (AI) metric (Pennec, 2006; Bhatia, 2009), Bures-Wasserstein (BW) metric (Bhatia et al., 2019), and Log-Euclidean (LE) metric (Arsigny et al., 2007) to endow $\mathrm{SPD}(m)$ with a Riemannian structure.

Let $\mathbf{W}, \widehat{\mathbf{W}} \in \mathrm{SPD}(m), \mathbf{U}, \mathbf{V} \in \mathrm{SYM}(m)$ and denote $\mathbf{C} \in \mathrm{SYM}(m)$ such that $\mathbf{C}_{ij} = 1$ if $i = j$ and $\mathbf{C}_{ij} = \frac{1}{\sqrt{2}}$ for $i \neq j$ and $c_{ij} = \mathbf{C}_{ij}$. $\mathbf{W} = \mathbf{PDP}^T$ is the eigenvalue decomposition of $\mathbf{W}$, where $\mathbf{P}, \mathbf{D} \in \mathbb{R}^{m \times m}$ and $\mathbf{D}$ is diagonal matrix of eigenvalues $[\lambda_1, \ldots, \lambda_m]$ and $\mathbf{P}$ is an orthogonal matrix. We denote the Hadamard between product two square matrices with $\odot$.

**SPD with Affine-Invariant metric.** The AI metric defined as $\langle \mathbf{U}, \mathbf{V} \rangle_{\mathbf{W}}^{\mathrm{AI}} := \mathrm{Tr}\left(\mathbf{W}^{-1}\mathbf{U}\mathbf{W}^{-1}\mathbf{V}\right)$. The reference point for Algorithm 2 is $\widehat{\mathbf{W}} = \mathbf{I}$ and the AI metric at $\mathbf{I}$ simplifies as $\langle \mathbf{U}, \mathbf{V} \rangle_{\widehat{\mathbf{W}}}^{\mathrm{AI}} = \mathrm{Tr}(\mathbf{UV})$. As the parallel transport operation is well-known for the AI metric, we choose it as the linear isometric transportation operation in Algorithm 2. The concrete implementation of Algorithm 2 for the SPD manifold with the AI metric is shown in Algorithm 3. The computational cost of implementing Algorithm 3 is $\mathcal{O}(m^3)$. Furthermore, the implicit basis that is being used by Algorithm 3 at $\mathbf{W}$ is

$$\mathscr{B}_{\mathbf{W}}^{\mathrm{AI}} = \left\{ c_{ij}.\mathbf{W}^{1/2}\left[\mathbf{e}_i\mathbf{e}_j^T + \mathbf{e}_j\mathbf{e}_i^T\right]\mathbf{W}^{1/2} : i = 1, \ldots, m, j = i+1, \ldots, m \right\},$$

where $\mathbf{W}^{1/2} = \mathbf{PD}^{1/2}\mathbf{P}$ denotes the principal square root of $\mathbf{W}$.

---

**Algorithm 3:** Sampling on SPD with Affine-Invariant metric

**Input** : Base point $\mathbf{W}$.
**Output:** Tangent Gaussian sample $\mathbf{U} \sim \mathcal{N}_{\mathbf{W}}(0, \sigma^2)$.
1 Generate normal random vector $\mathbf{a} \sim \mathcal{N}(\mathbf{0}, \sigma^2\mathbf{I}_{\frac{m(m+1)}{2}})$.
2 Reshape $\mathbf{a} \in \mathbb{R}^{m(m+1)/2}$ into $\mathbf{A} \in \mathrm{SYM}(m)$.
3 $\mathbf{U} = \mathrm{PT}^{\mathbf{I}_m \to \mathbf{W}}(\mathbf{C} \odot \mathbf{A})$.

---

**SPD with Bures-Wasserstein metric.** The BW metric is defined as $\langle \mathbf{U}, \mathbf{V} \rangle_{\mathbf{W}}^{\mathrm{BW}} := \mathrm{Tr}(\mathcal{L}_{\mathbf{W}}[\mathbf{U}]\mathbf{V})$, where $\mathcal{L}_{\mathbf{W}}[\mathbf{U}]$ is the solution to the matrix equation $\mathcal{L}_{\mathbf{W}}[\mathbf{U}]\mathbf{U} + \mathbf{U}\mathcal{L}_{\mathbf{W}}[\mathbf{U}] = \mathbf{U}$. The reference point for Algorithm 2 is $\widehat{\mathbf{W}} = \mathbf{I}$ and the BW metric at $\mathbf{I}$ simplifies as $\langle \mathbf{U}, \mathbf{V} \rangle_{\widehat{\mathbf{W}}}^{\mathrm{BW}} = \mathrm{Tr}(\mathbf{UV})/4$. We choose the parallel transport as the preferred isometric transportation procedure. The concrete implementation is shown in Algorithm 4 and the cost of implementation is $\mathcal{O}(m^3)$. The implicit basis at $\mathbf{W}$ that is being used by Algorithm 4 is

$$\mathscr{B}_{\mathbf{W}}^{\mathrm{BW}} = \left\{ c_{ij}\mathbf{P}\left[\mathbf{K} \odot \left(\mathbf{P}^T\left[\mathbf{e}_i\mathbf{e}_j^T + \mathbf{e}_j\mathbf{e}_i^T\right]\mathbf{P}\right)\right]\mathbf{P}^T : i = 1, \ldots, m, j = i+1, \ldots, m \right\},$$

where $\mathbf{K} \in \mathbb{R}^{m \times m}$ such that $\mathbf{K}_{rs} = \sqrt{\frac{\lambda_r + \lambda_s}{2}}$.

---

**Algorithm 4:** Sampling on SPD with Bures-Wasserstein metric

**Input** : Base point $\mathbf{W}$.
**Output:** Tangent Gaussian sample $\mathbf{U} \sim \mathcal{N}_{\mathbf{W}}(0, \sigma^2)$.
1 Generate a normal random vector $\mathbf{a} \sim \mathcal{N}(\mathbf{0}, \sigma^2\mathbf{I}_{\frac{m(m+1)}{2}})$.
2 Reshape $\mathbf{a} \in \mathbb{R}^{m(m+1)/2}$ into $\mathbf{A} \in \mathrm{SYM}(m)$.
3 $\mathbf{U} = \mathrm{PT}^{\mathbf{I}_m \to \mathbf{W}}(4\mathbf{C} \odot \mathbf{A})$.

---

**SPD with Log-Euclidean metric.** The LE metric is defined as $\langle \mathbf{U}, \mathbf{V} \rangle_{\mathbf{W}}^{\mathrm{LE}} :=$ $\mathrm{Tr}\left(\mathrm{DLogm}[\mathbf{W}](\mathbf{U})\mathrm{DLogm}[\mathbf{W}](\mathbf{V})\right)$, where $\mathrm{DLogm}[\mathbf{W}](\mathbf{U})$ is directional derivative of matrix logarithm of $\mathbf{W}$ evaluated at $\mathbf{U}$.

The reference point for Algorithm 2 is $\widehat{\mathbf{W}} = \mathbf{I}$ and the LE metric at $\mathbf{I}$ simplifies as $\langle \mathbf{U}, \mathbf{V} \rangle_{\widehat{\mathbf{W}}}^{\mathrm{LE}} = \mathrm{Tr}(\mathbf{UV})$. With the parallel transport as the preferred isometric transportation procedure, the sampling implementation is

shown in Algorithm 5. The computational cost of this implementation is $\mathcal{O}(m^3)$. The implicit basis that is being used by Algorithm 5 at $\mathbf{W}$ is

$$\mathscr{B}_{\mathbf{W}}^{\mathrm{LE}} = \left\{ c_{ij} \mathbf{P} \left[ \mathbf{K} \odot \left( \mathbf{P}^T \left[ \mathbf{e}_i \mathbf{e}_j^T + \mathbf{e}_j \mathbf{e}_i^T \right] \mathbf{P} \right) \right] \mathbf{P}^T : i = 1, \ldots, m, j = i+1, \ldots, m \right\},$$

where $\mathbf{K} \in \mathbb{R}^{m \times m}$ such that $\mathbf{K}_{rs} = f(\lambda_r, \lambda_s)$, where $f(x, y) = \frac{x-y}{\exp(x) - \exp(y)}$ if $x \neq y$ else $f(x, y) = \frac{1}{\exp(x)}$ when $x = y$.

---

**Algorithm 5:** Sampling on SPD with Log-Euclidean metric

    **Input** : Base point $\mathbf{W}$.
    **Output:** Tangent Gaussian sample $\mathbf{U} \sim \mathcal{N}_{\mathbf{W}}(0, \sigma^2)$.
**1** Generate a normal random vector $\mathbf{a} \sim \mathcal{N}(\mathbf{0}, \sigma^2 \mathbf{I}_{\frac{m(m+1)}{2}})$.
**2** Reshape $\mathbf{a} \in \mathbb{R}^{m(m+1)/2}$ into $\mathbf{A} \in \mathrm{SYM}(m)$.
**3** $\mathbf{U} = \mathrm{PT}^{\mathbf{I}_m \to \mathbf{X}}(\mathbf{C} \odot \mathbf{A})$.

---

### 4.2 Hyperbolic

We consider the two popular geometric models of the hyperbolic space: the Poincaré ball and the Lorentz hyperboloid model (Nickel & Kiela, 2017; 2018).

**Poincaré ball.** The Poincaré ball model is defined as $\mathrm{PB}(m) = \{\mathbf{w} \in \mathbb{R}^m : \|\mathbf{w}\|_2 < 1\}$ with the metric given by $\langle \mathbf{u}, \mathbf{v} \rangle_{\mathbf{w}}^{\mathrm{PB}} = 4\langle \mathbf{u}, \mathbf{v} \rangle_2 / (1 - \|\mathbf{w}\|_2^2)$. The tangent space at any $\mathbf{w} \in \mathrm{PB}(m)$ is $T_{\mathbf{w}}\mathrm{PB}(m) = \mathbb{R}^m$. The reference point for Algorithm 2 is $\widehat{\mathbf{w}} = \mathbf{o} \in \mathbb{R}^m$, where $\mathbf{o}$ denotes the zero vector. The PB metric at $\mathbf{o}$ is $\langle \mathbf{u}, \mathbf{v} \rangle_{\widehat{\mathbf{w}}}^{\mathrm{PB}} = \langle \mathbf{u}, \mathbf{v} \rangle_2$. The sampling algorithm is concretely shown in Algorithm 6 whose implementation cost is $\mathcal{O}(m)$. The implicit basis that is being used by Algorithm 6 is

$$\mathscr{B}_{\mathbf{x}}^{\mathrm{PB}} = \{\mathbf{e}_i (1 - \|\mathbf{w}\|_2^2)/4 : i = 1, \ldots, m\},$$

where $\mathbf{e}_i \in \mathbb{R}^m$, $i = 1, \ldots, m$ denotes the standard basis vector.

---

**Algorithm 6:** Sampling on Poincaré ball

    **Input** : Base point $\mathbf{w} \in \mathrm{PB}(m)$.
    **Output:** Tangent Gaussian sample $\mathbf{u} \sim \mathcal{N}_{\mathbf{w}}(0, \sigma^2)$.
**1** Generate a normal random vector $\mathbf{a} \sim \mathcal{N}(\mathbf{0}, \sigma^2 \mathbf{I}_m)$.
**2** $\mathbf{u} = \mathrm{PT}^{\mathbf{o}_m \to \mathbf{x}}(\mathbf{a}/4)$.

---

**Lorentz hyperboloid.** The Lorentizian inner product for $\mathbf{x}, \mathbf{w} \in \mathbb{R}^n$ is given by $\langle \mathbf{x}, \mathbf{w} \rangle_{\mathcal{L}} = -x_1 y_1 + \sum_{i=2}^k x_i y_i$. The Loretnz hyperboloid model is defined as $\mathrm{LH}(k) = \{\mathbf{w} \in \mathbb{R}^k | \langle \mathbf{w}, \mathbf{w} \rangle_{\mathcal{L}} = -1\}$ with the Lorentizian inner product as the Riemannian metric. The tangent space at $\mathbf{w} \in \mathrm{LH}(k)$ is given by $T_{\mathbf{w}}\mathrm{LH}(k) = \{\mathbf{u} \in \mathbb{R}^k | \langle \mathbf{w}, \mathbf{u} \rangle_{\mathcal{L}} = 0\}$. The reference point for Algorithm 2 is $\widehat{\mathbf{w}} = \mathbf{e}_1 \in \mathbb{R}^m$, the LH metric at $\mathbf{e}_1$ simplifies as $\langle \mathbf{u}, \mathbf{v} \rangle_{\widehat{\mathbf{w}}}^{\mathrm{LH}} = \langle \mathbf{u}, \mathbf{v} \rangle_2$ for $\mathbf{u}, \mathbf{v} \in T_{\widehat{\mathbf{w}}}\mathrm{PB}(m)$ and tangent space simplifies as $T_{\widehat{\mathbf{w}}}\mathrm{PB}(m) = \{0\} \times \mathbb{R}^{m-1}$. The sampling algorithm is concretely shown in Algorithm 7. The implementation cost is $\mathcal{O}(m)$. The implicit basis that is being used by Algorithm 7 at $\mathbf{w}$ is

$$\mathscr{B}_{\mathbf{w}}^{\mathrm{LH}} = \left\{ \bar{\mathbf{e}}_i - \frac{w_{i+1}}{1 + w_1} (\mathbf{e}_i + \mathbf{w}) : i = 1, \ldots, m-1 \right\},$$

where $\bar{\mathbf{e}}_i = (0, \widetilde{\mathbf{e}}_i)$ and $\mathbf{e}_i \in \mathbb{R}^m, \widetilde{\mathbf{e}}_i \in \mathbb{R}^{m-1}$ denotes standard basis vectors for $i = 1, \ldots, m-1$.

---

**Algorithm 7:** Sampling on Lorentz hyperboloid

    **Input** : Base point $\mathbf{w} \in \mathrm{PB}(m)$.
    **Output:** Tangent Gaussian sample $\mathbf{u} \sim \mathcal{N}_{\mathbf{w}}(0, \sigma^2)$.
**1** Generate a normal random vector $\mathbf{a} \sim \mathcal{N}(\mathbf{0}, \sigma^2 \mathbf{I}_{m-1})$.
**2** Perform zero padding $\mathbf{a} = [0, \mathbf{a}] \in \mathbb{R}^m$.
**3** $\mathbf{u} = \mathrm{PT}^{\mathbf{e}_1 \to \mathbf{w}}(\mathbf{a})$.

---

### 4.3 Sphere manifold

The sphere manifold is denoted as the set $\text{SP}(m) = \{\mathbf{w} \in \mathbb{R}^m | \|\mathbf{w}\|_2 = 1\}$ and the tangent space at $\mathbf{w}$ is given by $T_{\mathbf{w}}\text{SP}(m) = \{\mathbf{u} \in \mathbb{R}^m | \langle \mathbf{w}, \mathbf{u} \rangle_2 = 0\}$. The Riemannian metric is the induced by the Euclidean metric, i.e., $\langle \mathbf{u}, \mathbf{v} \rangle_{\mathbf{w}} = \langle \mathbf{u}, \mathbf{v} \rangle_2$. The reference point for Algorithm 2 is $\widehat{\mathbf{w}} = \mathbf{e}_1 \in \mathbb{R}^m$ and SP metric at $\mathbf{e}_1$ simplifies as $\langle \mathbf{u}, \mathbf{v} \rangle_{\widehat{\mathbf{w}}}^{\text{SP}} = \langle \mathbf{u}, \mathbf{v} \rangle_2$. With parallel transport as the preferred isometric transportation procedure, the sampling implementation is shown in Algorithm 8 with implementation cost $\mathcal{O}(m)$. The implicit basis being used by Algorithm 8 is

$$\mathscr{B}_{\mathbf{w}}^{\text{SP}} = \{\bar{\mathbf{e}}_i - w_{i+1} \cdot \mathbf{w} : i = 1, \ldots, m-1\},$$

where $\bar{\mathbf{e}}_i = (0, \widetilde{\mathbf{e}}_i)$ and $\widetilde{\mathbf{e}}_i \in \mathbb{R}^{m-1}$ denotes the standard basis vector for $i = 1, \ldots, m-1$.

---

**Algorithm 8:** Sampling on sphere

    **Input** : Base point $\mathbf{w} \in \text{SP}(m)$.
    **Output:** Tangent Gaussian sample $\mathbf{u} \sim \mathcal{N}_{\mathbf{w}}(0, \sigma^2)$.
**1** Generate a normal random vector $\mathbf{a} \sim \mathcal{N}(\mathbf{0}, \sigma^2 \mathbf{I}_{m-1})$.
**2** Perform zero padding $\mathbf{a} = [0, \mathbf{a}] \in \mathbb{R}^m$.
**3** $\mathbf{u} = \text{PT}^{\mathbf{e}_1 \to \mathbf{w}}(\mathbf{a})$.

---

### 4.4 Stiefel manifold

The Stiefel manifold is the set of column orthonormal matrices, i.e., $\text{ST}(m, r) = \{\mathbf{W} \in \mathbb{R}^{m \times r} | \mathbf{W}^T \mathbf{W} = \mathbf{I}\}$ and its tangent space at $\mathbf{W}$ is $T_{\mathbf{W}}\text{ST}(m, r) = \{\mathbf{U} \in \mathbb{R}^{m \times r} | \mathbf{U}^T \mathbf{W} + \mathbf{W}^T \mathbf{U} = \mathbf{O}\}$, where $\mathbf{O} \in \mathbb{R}^{r \times r}$ is the zero matrix. The Riemannian metric is the induced by the Euclidean metric $\langle \mathbf{U}, \mathbf{V} \rangle_{\mathbf{W}}^{\text{ST}} = \text{Tr}(\mathbf{U}^T \mathbf{V})$ (Edelman et al., 1998). The reference point for Algorithm 2 is $\widehat{\mathbf{W}} = [\mathbf{e}_1, \ldots \mathbf{e}_r] \in \mathbb{R}^{m \times r}$, where the metric is $\langle \mathbf{U}, \mathbf{V} \rangle_{\widehat{\mathbf{W}}}^{\text{ST}} = \text{Tr}(\mathbf{U}^T \mathbf{V})$ and the tangent space is $T_{\widehat{\mathbf{W}}}\text{ST}(m, r) = \text{SKEW}(r) \times \mathbb{R}^{(m-r) \times r}$, where we denote $\text{SKEW}(r)$ as the set of skew-symmetric matrices of size $r \times r$. Huang et al. (2017) have proposed an efficient isometric vector transport procedure, which we choose for implementing Algorithm 2. The concrete sampling procedure is shown in Algorithm 9 with an implementation cost $\mathcal{O}(mr^2)$. The implicit basis that is being used by Algorithm 9 is

$$\mathscr{B}_{\mathbf{W}}^{\text{ST}} = \{\frac{1}{\sqrt{2}} \mathbf{W}(\mathbf{e}_i \mathbf{e}_j^T - \mathbf{e}_j \mathbf{e}_i^T) : i = 1 \ldots r, j = i+1, \ldots, r\} \cup \{\mathbf{W}_{\perp} \widetilde{\mathbf{e}}_i \mathbf{e}_j^T : i = 1, \ldots, m-r, j = 1, \ldots, r\},$$

where $\mathbf{e}_i \in \mathbb{R}^r, \widetilde{\mathbf{e}}_i \in \mathbb{R}^{m-r}$ denotes the standard basis vectors for $i = 1, \ldots, m-1$ and $\mathbf{W}_{\perp} \in \mathbb{R}^{m \times (m-r)}$ denotes a matrix such that the columns form an orthonormal basis of the orthogonal complement of the columns of $\mathbf{W}$.

---

**Algorithm 9:** Sampling on Stiefel manifold

    **Input** : Base point $\mathbf{W} \in \text{ST}(m)$.
    **Output:** Tangent Gaussian sample, $\mathbf{U} \sim \mathcal{N}_{\mathbf{W}}(0, \sigma^2)$.
**1** Generate a normal random vector $\mathbf{a}_1 \sim \mathcal{N}(\mathbf{0}, \sigma^2 \mathbf{I}_{\frac{r(r-1)}{2}})$ and reshape into $\mathbf{A}_1 \in \text{SKEW}(r)$.
**2** Generate a normal random vector $\mathbf{a}_2 \sim \mathcal{N}(\mathbf{0}, \sigma^2 \mathbf{I}_{(m-r) \times r})$ and reshape into $\mathbf{A}_2 \in \mathbb{R}^{(m-r) \times r}$.
**3** $\widehat{\mathbf{W}} = [\mathbf{e}_1, \ldots, \mathbf{e}_r], \mathbf{A} = \begin{bmatrix} \mathbf{A}_1/\sqrt{2} \\ \mathbf{A}_2 \end{bmatrix} \in \mathbb{R}^{m \times r}$.
**4** $\mathbf{U} = \text{VT}^{\widehat{\mathbf{W}} \to \mathbf{W}}(\mathbf{A})$.

---

### 4.5 Grassmann manifold

The Grassmann manifold $\text{GR}(m, r)$ consists of $r$-dimensional linear subspaces of $\mathbb{R}^m$ ($r \leq m$) and is represented as $\text{GR}(m, r) = \{\text{colspan}(\mathbf{W}) | \mathbf{W} \in \mathbb{R}^{m \times r}, \mathbf{W}^T \mathbf{W} = \mathbf{I}_r\}$, where colspan denotes the column space.

The tangent space at $\mathbf{W}$ is $T_{\mathbf{W}}\text{GR}(m,r) = \{\mathbf{U} \in \mathbb{R}^{m \times r} | \mathbf{U} \in \mathbb{R}^{m \times r}, \mathbf{W}^T \mathbf{U} = \mathbf{O}_r\}$ where $\mathbf{O}_r \in \mathbb{R}^{r \times r}$ is zero matrix (Edelman et al., 1998). The Riemannian metric is induced by the Euclidean metric $\langle \mathbf{U}, \mathbf{V} \rangle_{\mathbf{W}}^{\text{GR}} = \text{Tr}[\mathbf{U}^T \mathbf{V}]$ for $\mathbf{U}, \mathbf{V} \in T_{\mathbf{W}}\text{GR}(m,r)$. (Edelman et al., 1998). The reference point for Algorithm 2 is $\widehat{\mathbf{W}} = [\mathbf{e}_1, \ldots \mathbf{e}_r] \in \mathbb{R}^{m \times r}$, the metric at $\widehat{\mathbf{W}}$ is $\langle \mathbf{U}, \mathbf{V} \rangle_{\widehat{\mathbf{w}}}^{\text{GR}} = \text{Tr}(\mathbf{U}^T \mathbf{V})$ and tangent space at $\widehat{\mathbf{W}}$ simplifies as $T_{\widehat{\mathbf{W}}}\text{GR}(m,r) = \{0\}^{r \times r} \times \mathbb{R}^{(m-r) \times r}$, where we denote $\{0\}^{r \times r}$ as the singleton set of zero matrix of size $r \times r$. Similar to the Stiefel case, we use the vector transport may as the preferred isometric transportation procedure (Huang et al., 2017). The sampling implementation is shown in Algorithm 10 with computational cost $\mathcal{O}(mr^2)$. The implicit basis that is being used by Algorithm 10 is

$$\mathscr{B}_{\mathbf{W}}^{\text{GR}} = \{\mathbf{W}_\perp \widetilde{\mathbf{e}}_i \mathbf{e}_j^T : i = 1, \ldots, m-r, j = 1, \ldots, r\},$$

where $\mathbf{e}_i \in \mathbb{R}^r, \widetilde{\mathbf{e}}_i \in \mathbb{R}^{m-1}$ denote the standard basis vectors for $i = 1, \ldots, m-1$ and $\mathbf{W}_\perp \in \mathbb{R}^{m \times (m-r)}$ denotes a matrix such that the columns form an orthonormal basis of the orthogonal complement of the column space of $\mathbf{W}$.

---

**Algorithm 10:** Sampling on Grassmann manifold

    **Input** : Base point $\mathbf{W} \in \text{GR}(m)$.
    **Output:** Tangent Gaussian sample, $\mathbf{U} \sim \mathcal{N}_{\mathbf{W}}(0, \sigma^2)$.
**1** Generate a normal random vector $\mathbf{a} \sim \mathcal{N}(\mathbf{0}, \sigma^2 \mathbf{I}_{(m-r) \times r})$ and reshape into $\mathbf{A} \in \mathbb{R}^{(m-r) \times r}$.
**2** $\widehat{\mathbf{W}} = [\mathbf{e}_1, \ldots, \mathbf{e}_r]$ , $\mathbf{A} = \begin{bmatrix} \mathbf{O}_r \\ \mathbf{A} \end{bmatrix} \in \mathbb{R}^{m \times r}$.
**3** $\mathbf{U} = \text{VT}^{\widehat{\mathbf{W}} \to \mathbf{W}}(\mathbf{A})$.

---

# 5 Private Riemannian variance reduced stochastic optimization

Variance reduced stochastic optimization methods (Roux et al., 2012; Johnson & Zhang, 2013; Defazio et al., 2014; Reddi et al., 2016) employ a hybrid update rule that uses both full gradient and stochastic gradient information simultaneously. By doing so, variance reduced methods improve the gradient complexity compared to the stochastic and the full gradient descent methods by requiring less gradient calls to achieve the same convergence rates than the full gradient descent method. Many variance reduction strategies that work in the Euclidean space have also been generalized to manifolds (Zhang et al., 2016; Sato et al., 2019; Zhou et al., 2019; Han & Gao, 2021).

In this section, we privatize the Riemannian stochastic variance reduced gradient (RSVRG) algorithm (Zhang et al., 2016) for solving (1) and develop a differentially private RSVRG algorithm, henceforth denoted by DP-RSVRG. Our proposed DP-RSVRG is summarized in Algorithm 11. DP-RSVRG with restart is presented as Algorithm 12.

DP-RSVRG has two loops. In the inner loop, an unbiased variance reduced stochastic gradient is constructed by correcting the Riemannian stochastic gradient with the full gradient calculated at the outer loop. We add noise from the tangent Gaussian distribution to the variance reduced gradient. The clipping operation $\text{clip}_\tau : T_w \mathcal{M} \to T_w \mathcal{M}$ is defined as $\text{clip}_\tau(\nu) = \min \left\{ \frac{\|\nu\|_w}{\tau}, 1 \right\} \nu$ and it ensures that the norm of $\nu$ is at most $\tau$. The norm of the full gradient is clipped with parameter $\mathcal{C}_0$ and the variance reduced gradient with parameter $\mathcal{C}_1$, respectively. PT refers to the parallel transport operation.

## 5.1 Privacy guarantee

In this section, we analyze the privacy guarantees of DP-RSVRG. We begin by noting that the variance reduced stochastic gradient has a deterministic and a subsampled component. Hence, Step 7 of Algorithm 11 can be equivalently re-written as

$$v_t^{s+1} = \text{clip}_{\mathcal{C}_1}\left(\text{grad}\, f(w_t^{s+1}; z_{i_t})\right) - \text{PT}^{\widetilde{w}^s \to w_t^{s+1}}\left(\text{clip}_{\mathcal{C}_1}\left(\text{grad}\, f(\widetilde{w}^s; z_{i_t})\right) - (g^{s+1} + \xi_{t1}^s)\right) + \xi_{t2}^s, \qquad (2)$$

---

**Algorithm 11:** DP-RSVRG

    **Input** : update frequency $m$, learning rate $\eta$, number of epochs $S$, clipping parameters $\mathcal{C}_0, \mathcal{C}_1$, and initial iterate $w^0$.

**1** initialize $\widetilde{w} = w^0$.

**2** for $s = 0, 1, \ldots, S-1$ do

**3**     $w_0^{s+1} = \widetilde{w}^s$.

**4**     $g^{s+1} = \frac{1}{n} \sum_{i=1}^n \mathrm{clip}_{\mathcal{C}_0} (\mathrm{grad}\, f(\widetilde{w}^s; z_i))$.

**5**     for $t = 0, 1, \ldots, m-1$ do

**6**        Randomly pick $i_t \in \{1, \ldots, n\}$.

**7**        $v_t^{s+1} = \mathrm{clip}_{\mathcal{C}_1} (\mathrm{grad}\, f(w_t^{s+1}; z_{i_t})) - \mathrm{PT}^{\widetilde{w}^s \to w_t^{s+1}} \left( \mathrm{clip}_{\mathcal{C}_1} (\mathrm{grad}\, f(\widetilde{w}^s; z_{i_t})) - g^{s+1} \right) + \epsilon_t^{s+1}$, where $\epsilon_t^{s+1} \sim \mathcal{N}_{w_t^{s+1}}(0, \sigma^2)$.

**8**        $w_{t+1}^{s+1} = \mathrm{Exp}_{w_t^{s+1}}(-\eta v_t^{s+1})$.

**9**     Set $\widetilde{w}_a = w_m^{s+1}$.

**10 Output I** : $w^{\mathrm{priv}} = \widetilde{w}^S$.

**11 Output II** : $w^{\mathrm{priv}}$ is choosen uniformly randomly from $\{\{w_t^{s+1}\}_{t=0}^{m-1}\}_{s=0}^{S-1}$.

---

**Algorithm 12:** DP-RSVRG with restarts

    **Input** : update frequency $m$, learning rate $\eta$, number of epochs $S$, and initial iterate $w^0$.

**1** for $k = 0, 1, \ldots, K-1$ do

**2**     $w^{k+1} = \mathrm{DP\text{-}RSVRG}(m, \eta, S, w^k)$ with output option **II**.

---

where $\xi_{t1}^s \sim \mathcal{N}_{\widetilde{w}^s}(0, \sigma_1^2)$ and $\xi_{t2}^s \sim \mathcal{N}_{w_t^{s+1}}(0, \sigma_2^2)$. Specifically, the noise variance $\sigma^2$ is split into into $\sigma_1^2$ for the full gradient query and $\sigma_2^2$ for the variance reduced stochastic gradient query such that $\sigma_1^2 + \sigma_2^2 = \sigma^2$.

Claim 2 ensures that $\mathrm{PT}^{\widetilde{w}^s \to w_t^{s+1}} \xi_{t1}^s + \xi_{t2}^{s+1} = \epsilon_t^{s+1} \sim \mathcal{N}_{w_t^{s+1}}(0, \sigma^2)$. Hence, (2) can be viewed as a composition of a full gradient tangent Gaussian mechanism

$$\mathcal{R}^s(Z) = r^{s+1} = \frac{1}{n} \sum_{i=1}^n \mathrm{clip}_{\mathcal{C}_0} (\mathrm{grad}\, f(\widetilde{w}^s; z_i)) + \xi_{t1}^s,$$

where $\xi_{t1}^s \sim \mathcal{N}_{\widetilde{w}^s}(0, \sigma_1^2)$ and a variance reduced Gaussian mechanism

$$\mathcal{R}_t^{s+1}(Z) = \mathrm{clip}_{\mathcal{C}_1} (\mathrm{grad}\, f(w_t^{s+1}; z_{i_t})) - \mathrm{PT}^{\widetilde{w}^s \to w_t^{s+1}} \left( \mathrm{clip}_{\mathcal{C}_1} (\mathrm{grad}\, f(\widetilde{w}^s; z_{i_t})) - r^{s+1} \right) + \xi_{t2}^{s+1},$$

where $\xi_{t2}^{s+1} \sim \mathcal{N}_{w_t^{s+1}}(0, \sigma_2^2)$. We now prove the moments bounds on the full gradient mechanism $\mathcal{K}_{\mathcal{R}^s}$ and variance reduced mechanism $\mathcal{K}_{\mathcal{R}_t^{s+1}}$ in the following claims and the proofs are given in Section B.3.1.

**Claim 3.** *The moments bounds satisfy*

$$\mathcal{K}_{\mathcal{R}^s}(\lambda) \leq \frac{2\lambda(\lambda+1)\mathcal{C}_0^2}{n^2 \sigma_1^2} \text{ and } \mathcal{K}_{\mathcal{R}_t^{s+1}}(\lambda) \leq \frac{8\lambda(\lambda+1)\mathcal{C}_1^2}{\sigma_2^2}.$$

Now we derive the moments bound on subsampled version of $\mathcal{R}_t^{s+1}$ using the results given in (Wang et al., 2019b;c) and the proof is given in Section B.3.2.

**Claim 4.** *Define* subsample : $\mathcal{Z}^n \to \mathcal{Z}$ *as the process of sampling a single data point from $n$ data points uniformly randomly. Define the subsampled mechanism for $\mathcal{R}_t^{s+1}$ as $^{\mathrm{sub}}\mathcal{R}_t^{s+1} = \mathcal{R}_t^{s+1} \circ$ subsample. Suppose $\sigma_2 \geq 12\mathcal{C}_1^2$ and $\lambda \leq 2/3\sigma_2^2 \log\left(n(\lambda+1)(1 + (\sigma_2^2/16\mathcal{C}_1^2))\right)$, we have*

$$\mathcal{K}_{^{\mathrm{sub}}\mathcal{R}_t^{s+1}}(\lambda) \leq \frac{28\lambda(\lambda+1)\mathcal{C}_1^2}{n^2 \sigma_2^2}.$$

The full mechanism $\mathcal{R}$ can be seen as an adaptive composition of $\{\{\mathcal{K}_{\mathrm{sub}_{\mathcal{R}_t^{s+1}}}\}_{t=0}^{m-1}\}_{s=0}^{S-1}$ and $\{\{\mathcal{K}_{\mathcal{R}^s}\}_{t=0}^{m-1}\}_{s=0}^{S-1}$. Since $\sigma_1^2 + \sigma_2^2 = \sigma^2$, we can rewrite $\sigma_1^2 = \alpha\sigma^2$, $\sigma_2^2 = (1-\alpha)\sigma^2$ for some $\alpha \in (0,1)$. Using this claim, minimizing over $\alpha$, and setting $\mathcal{C} = \max\{\mathcal{C}_0, \mathcal{C}_1\}$, we have

$$\mathcal{K}_{\mathcal{R}}(\lambda) \leq \sum_{t=0}^{m}\sum_{s=0}^{S-1} \mathcal{K}_{\mathrm{sub}_{\mathcal{R}_t^{s+1}}}(\lambda) + \sum_{t=0}^{m}\sum_{s=0}^{S-1} \mathcal{K}_{\mathcal{R}^{s+1}}(\lambda) \leq \frac{2mS\lambda(\lambda+1)\mathcal{C}_0^2}{n^2\sigma_1^2} + \frac{28mS\lambda(\lambda+1)\mathcal{C}_1^2}{n^2\sigma_2^2}$$

$$\Rightarrow \mathcal{K}_{\mathcal{R}}(\lambda) \leq \min_{\alpha\in(0,1)} \frac{mS\lambda(\lambda+1)\mathcal{C}^2}{n^2\sigma^2}\left[\frac{2}{\alpha} + \frac{28}{1-\alpha}\right]. \tag{3}$$

It should be noted that for a given $\lambda$, the minimization over $\alpha$ has a closed-form solution.

The moments bound $\mathcal{K}_{\mathcal{R}}$ given in (3) can be converted to $(\epsilon, \delta)$ guarantee using conversion rules, e.g., based on (Mironov, 2017, Proposition 3): Given $0 < \delta < 1$, $\epsilon = \min_{\lambda \geq 1} \frac{\mathcal{K}_{\mathcal{R}}(\lambda-1) + \log 1/\delta}{\lambda-1}$. Recently, however, the optimal conversion rule has been given in (Asoodeh et al., 2020, Theorem 3) for which there exists no closed-form expression but can be solved numerically to get $\epsilon$. The solver is available in the `autodp` library (Wang et al., 2019c). The above result connecting the moment bound $\mathcal{K}_{\mathcal{R}}$ with $\alpha$ in (3) implies that tighter $(\epsilon, \delta)$ guarantees can be obtained by optimizing over $\alpha$, i.e., by exploiting the inter-play between the the noise added to the full gradient and that to the variance reduced gradient.

It should be emphasized that in the Euclidean setting, Wang et al. (2017) have not considered optimization of $\alpha$ as in (3). We empirically show that such an optimization of $\alpha$ obtains significant improvement in privacy in Section 6.2. We end this section with the following privacy result for Algorithms 11 and 12.

**Claim 5.** *Algorithms 11 and 12 are $(\epsilon, \delta)$-differentially private with $\sigma^2 \geq c_1 \frac{mS\log(1/\delta)\mathcal{C}^2}{n^2\epsilon^2}$ and $\sigma^2 \geq c_2 \frac{mSK\log(1/\delta)\mathcal{C}^2}{n^2\epsilon^2}$, respectively, for some positive constants $c_1, c_2$ and $\mathcal{C} = \max\{\mathcal{C}_0, \mathcal{C}_1\}$.*

### 5.2 Utility guarantee

In this section, we prove the utility guarantees of DP-RSVRG under various function classes on manifolds including geodesic strong convex functions, general nonconvex functions, and functions that satisfy the Riemannian Polyak–Łojasiewicz (PL) condition. In particular, the geodesic strong convexity and Riemannian PL conditions generalize the notions of strong convexity and PL condition from the Euclidean space to manifolds, allowing fast convergence (for problems satisfying these conditions) to global optimality when optimizing on manifolds. The proofs of the results discussed in this section are included in Sections B.4.1-B.4.3.

Let $\mathcal{W} \subseteq \mathcal{M}$ be a totally normal neighborhood and $D_{\mathcal{W}}$ denotes its diameter and $\kappa_{\min}$ is the lower bound on curvature of $\mathcal{W}$ (discussed in details in Section 2). Following (Zhang & Sra, 2016; Han & Gao, 2021; Han et al., 2022a), we make the below standard assumption.

**Assumption 1.** *Each $f_i$ in (1) is $L$-geodesically smooth and $L_0$-geodesically Lipschitz over $\mathcal{W}$.*

The gradient complexity of an algorithm is measured in the number of incremental first-order oracle (IFO) calls needed. An IFO (Agarwal & Bottou, 2015) takes an index $i \in [n]$, $w \in \mathcal{W}$ and returns $(f_i(w), \mathrm{grad}\, f_i(w)) \in \mathbb{R} \times T_w\mathcal{M}$. Also, for readability we hide the log factors through notation $\widetilde{\mathcal{O}}$ in the utility bounds and gradient complexities. The exact expressions are in (11), (12) for $\mu$-strongly convex functions; (16), (17) for non-convex functions; and (18), (19) for functions with the Riemannian PL condition in the appendix section.

**Theorem 6** (Utility under geodesic strong convexity). *Suppose that Assumption 1 holds and $F$ is $\mu$-strongly geodesic convex over $\mathcal{W}$. If we run the Algorithm 11 with learning rate $\eta = \mathcal{O}(\frac{\mu}{\zeta L^2})$, frequency $m = \widetilde{O}(\frac{\zeta L^2}{\mu^2})$ for $S = \mathcal{O}(\log(\frac{n\epsilon\mu}{\log(1/\delta)\zeta L_0^2 d}))$ outer loops with output **I**, then $\mathbb{E}[F(w^{\mathrm{priv}}) - F(w^*)] = \widetilde{\mathcal{O}}\left(\frac{d\zeta LL_0^2 \log(1/\delta)\mathbb{E}[\mathrm{dist}^2(w^0, w^*)]}{\mu^2 n^2\epsilon^2}\right)$. Furthermore, the gradient complexity is given by $\widetilde{\mathcal{O}}(n + \frac{\zeta L^2}{\mu^2})$.*

**Theorem 7** (Utility under nonconvex functions). *Suppose that Assumption 1 holds. If we run the Algorithm 11 with output **II**, learning rate $\eta = \mathcal{O}(\frac{1}{Ln^{2/3}\zeta^{1/2}})$, frequency $m = \Theta(n)$ and for $S = \sqrt{\frac{L\zeta}{d\log(1/\delta)}}\frac{n^{2/3}\epsilon}{L_0}$*

*outer loops, then* $\mathbb{E}\|\operatorname{grad} F(w^{\mathrm{priv}})\|^2 \leq \frac{L_0\sqrt{dL\log(1/\delta)\mathbb{E}[F(w^0)-F(w^*)]}}{n\epsilon}$. *The gradient complexity is given by* $\mathcal{O}(\sqrt{\frac{L\zeta}{d\log(1/\delta)}}\frac{n^{5/3}\epsilon}{L_0})$.

We now use Algorithm 12 to achieve utility guarantee under the Riemannian PL condition.

**Theorem 8** (Utility under Riemannian PL condition)**.** *Suppose that Assumption 1 holds and* $F = \frac{1}{n}\sum_{i=1}^n f_i(w)$ *satisfies the Riemannian PL condition with parameter* $\tau$. *If we run Algorithm 12 with learning rate* $\eta = \mathcal{O}(\frac{1}{Ln^{2/3}\zeta^{1/2}})$, *frequency* $m = \Theta(n)$, $S = \mathcal{O}(1)$, *and* $K = \log(\frac{n^2\epsilon^2}{dL\tau^2\log(1/\delta)L_0^2})$, *then* $\mathbb{E}[F(w^{\mathrm{priv}}) - F(w^*)] \leq \widetilde{\mathcal{O}}(\frac{dL\tau^2\log(1/\delta)L_0^2}{n^2\epsilon^2})$. *Furthermore, the gradient complexity is given by* $\widetilde{\mathcal{O}}(L\tau\zeta^{1/2}n^{2/3})$.

### 5.3 Discussion: DP-RGD vs DP-RSGD vs DP-RSVRG

In this section, we compare our proposed DP-RSVRG with DP-RGD (Han et al., 2022a) and DP-RSGD (Han et al., 2022a).

1. **Strongly geodesic convex**: DP-RSGD and DP-RGD both assume $f_i$ in (1) to be $\mu-$strongly g-convex. In contrast, DP-RSVRG in Theorem 6 just assumes that $F = \sum_{i=1}^n f_i$ to be $\mu$-strongly *g*-convex, which is a much weaker assumption. Furthermore, DP-RSVRG assumes $f_i$ to be $L$-g-smooth, while DP-RGD, DP-RSGD do not make any smoothness assumption.

   For $\mu$-strongly g-convex functions, DP-RGD and DP-RSGD obtain the utility bound $\mathcal{O}\left(\frac{d\zeta L_0^2 \log(1/\delta)\mathbb{E}[\mathrm{dist}^2(w_0,w^*)]}{\mu n^2\epsilon^2}\right)$ with gradient complexities $n^2$ and $n^3$, respectively (Han et al., 2022a, Theorem 3). On the other hand, DP-RSVRG obtains a utility bound $\widetilde{\mathcal{O}}\left(\frac{d\zeta LL_0^2\log(1/\delta)\mathbb{E}[\mathrm{dist}^2(w_S^0,w^*)]}{\mu^2 n^2\epsilon^2}\right)$ in $\widetilde{\mathcal{O}}(n + \frac{\zeta L^2}{\mu^2})$ IFO calls. DP-RSVRG bounds are worse in terms of condition number $L/\mu$ due to the weaker assumption as discussed above.

2. **Riemannian PL condition**: DP-RSGD and DP-RGD both assume $f_i$ in (1) to satisfy the Riemannian PL condition with parameter $\tau$. On the other hand, DP-RSVRG in Theorem 8 assumes that $F = \sum_{i=1}^n f_i$ satisfies the same condition, which is weaker.

   DP-RGD and DP-RSGD obtain utility bound of $\widetilde{\mathcal{O}}\left(\frac{\tau^{-1}d\log(1/\delta)L_0^2\mathbb{E}[F(w^0)-F(w^*)]}{n^2\epsilon^2}\right)$ in $n\log\left(\frac{n^2\epsilon^2}{dL_0^2\log(1/\delta)}\right)$ and $\log\left(\frac{n^2\epsilon^2}{dL_0^2\log(1/\delta)}\right)$ IFO calls, respectively. DP-RSVRG obtains a utility bound $\widetilde{\mathcal{O}}\left(\frac{dL\tau^2\log(1/\delta)L_0^2}{n^2\epsilon^2}\right)$ in $\widetilde{\mathcal{O}}(L\tau\zeta^{1/2}n^{2/3})$ IFO calls. DP-RSVRG bounds are worse in terms of PL parameter $\tau$ because of weaker assumption as mentioned above.

3. **Nonconvex**: In the nonconvex setting, only a bound on the gradient norm can be obtained instead of a bound on the excess risk. Both DP-RGD and DP-RSGD obtain bound on gradient norm as $\mathcal{O}(\frac{L_0\sqrt{dL\log(1/\delta)}}{n\epsilon})$ in $\mathcal{O}(\frac{\sqrt{L}n^2\epsilon}{L_0\sqrt{d\log(1/\delta)}})$ and $\mathcal{O}(\frac{\sqrt{L}n\epsilon}{L_0\sqrt{d\log(1/\delta)}})$ iterations respectively (Han et al., 2022a, Theorem 5). From Theorem 7, DP-RSVRG obtains bound on gradient as $\mathcal{O}(\frac{L_0\sqrt{dL\log(1/\delta)}}{n\epsilon})$ in $\mathcal{O}(\sqrt{\frac{L\zeta}{d\log(1/\delta)}}\frac{n^{5/3}\epsilon}{L_0})$ iterations. Hence, in this case, DP-RGD, DP-RSGD, and DP-RSVRG have the same matching utility bounds.

## 6 Experiments

In this section, we illustrate the efficacy of the proposed sampling procedures and the proposed DP-RSVRG algorithm. We also show the benefit of $\alpha$ optimization (Section 5.1) in terms of the gain in privacy guarantee.

### 6.1 Benchmarking of different sampling procedures

We benchmark our proposed isometric transportation (Algorithm 2) based sampling, denoted as 'Transportation', with the following three baselines.

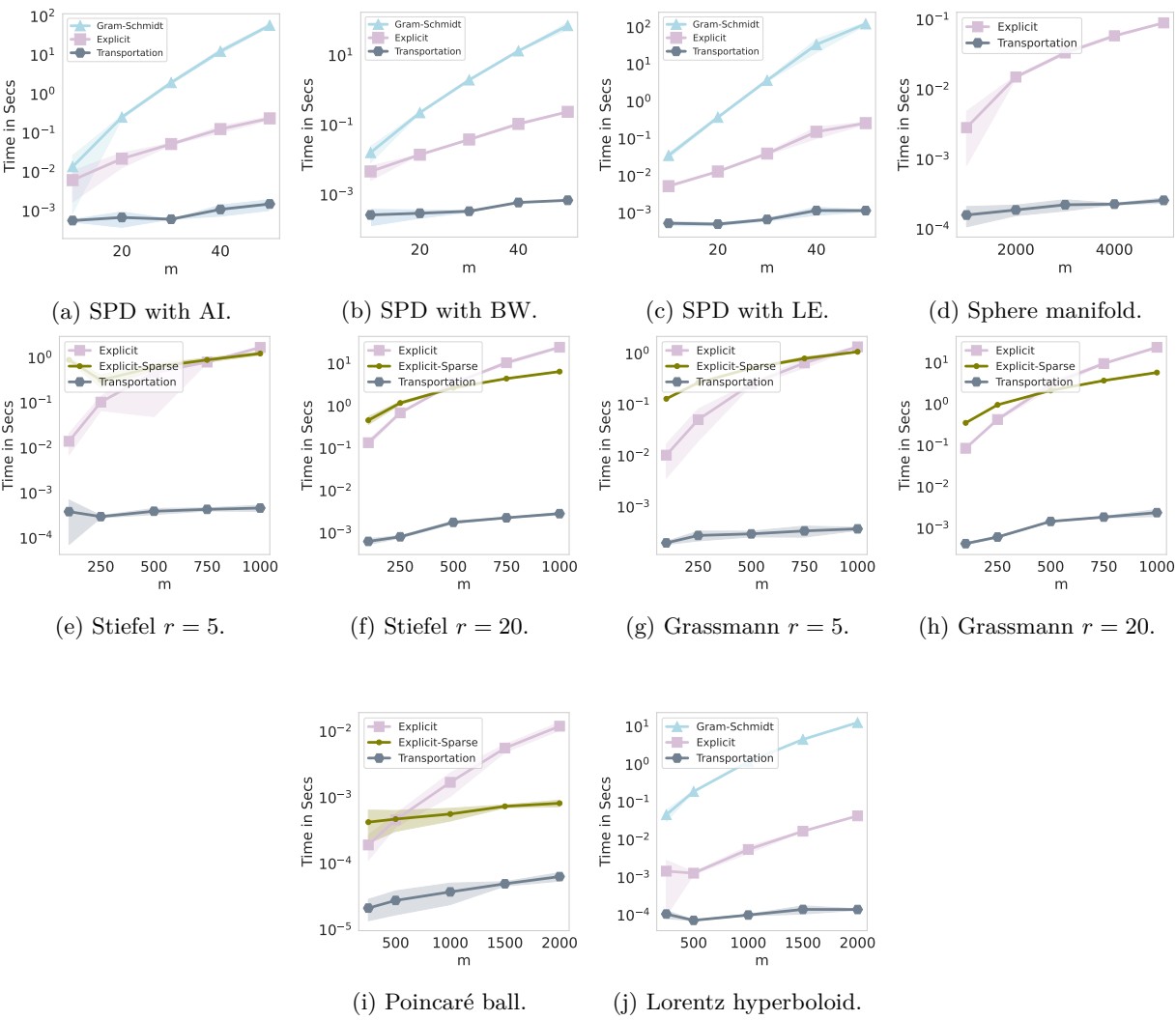

Figure 1: Benchmarking of different sampling strategies. As can be seen, our proposed method 'Transportation' consistently outperforms the other baselines on the manifolds.

1. **Sampling using Gram-Schmidt.** We perform Gram-Schmidt orthogonalization on the basis $\mathscr{B}$ that is orthonormal wrt the Euclidean metric. This is a baseline for the SPD and Lorentz hyperboloid manifolds because for other manifolds, the orthonormal basis with respect to the underlying metric $\langle .,. \rangle_w$ can be simply obtained by scaling $\mathscr{B}$. This is denoted as 'Gram-Schmidt'.

2. **Sampling using explicit basis.** We take the implicit bases generated by the isometric transportation strategy (Algorithm 2) and generate them explicitly, i.e., construct the full basis and then perform linear combinations. This is denoted as 'Explicit'.

3. **Sampling using explicit basis by exploiting sparsity.** As an additional baseline, we implement sampling with explicit basis construction using sparse operations. Sparsity is present in Stiefel, Grassmann, and Poincaré ball bases, and is therefore a baseline only for these three manifolds. This is denoted as 'Explicit-Sparse'.

In Figure 1, we benchmark the sampling time for generating a single sample from the tangent Gaussian distribution on various manifolds discussed in Section 4. For $\text{SPD}(m)$, we consider $m = \{5, 10, 20, 30, 50\}$.

Table 2: Overhead of privatizations for DP-RSGD (with $3 \times 10^5$ epochs) for the SPD Fréchet mean and the principal eigenvector problems. Our proposed isometric transportation based sampling strategy lead to orders of magnitude improvements than those of Han et al. (2022a).

| Manifold | Size | Han et al. (2022a) | This work |
|---|---|---|---|
| SPD | $11 \times 11$ | 660 hrs | 41 seconds ($\sim 10^4$ improvement) |
| Sphere | 786 | 668 seconds | 24 seconds ($\sim 10$ improvement) |

For PB($m$), LH($m$), and SP($m$), we consider $m = \{250, 500, 1000, 1500, 2000\}$. For GR($m, r$) and ST($m, r$), we consider $m = \{100, 250, 500, 750, 1000\}$ and $r = \{10, 20\}$.

Figure 1 shows the average sampling time over five different base points chosen at random. From the figure, we see that the transportation sampling strategy is faster by two to four orders of magnitude than all the considered baselines. It also shows the benefit of the transportation strategy as a unified sampling framework.

We study the benefits of the proposed sampling procedures in two problems: private estimation of the SPD Fréchet mean and the principal eigenvector (discussed in Section 6.3). We use DP-RSGD algorithm for both problems and compare our sampling strategy with that developed in (Han et al., 2022a). The results are shown in Table 2. We observe that the proposed sampling strategy offers significant improvements leading to minimal overhead due to privatization.

## 6.2 Optimizing $\alpha$ in moments bound for better $(\epsilon, \delta)$ guarantees

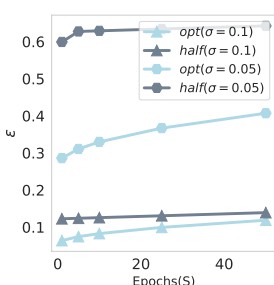

We now show that better privacy guarantees can be empirically achieved by optimizing $\alpha$ in moments bound (Section 5.1). To this end, we use the `autodp` library (Wang et al., 2019c) and set $\sigma_1 = \sqrt{\alpha}\sigma, \sigma_2 = \sqrt{(1-\alpha)}\sigma$ instead of the standard setting $\sigma_1 = \sigma_2 = \sigma/\sqrt{2}$. We fix $\mathcal{C}_1 = 0.1, \mathcal{C}_2 = 0.01$ and frequency to $m = 10000$ and $n = 100000$. The results are shown in Figure 2 for epochs $S = \{1, 5, 10, 25, 50, 100\}$ and noise $\sigma = \{0.1, 0.05\}$. We observe that our proposal to optimize over $\alpha$ significantly improves the privacy guarantees than the standard setting. For noise level $\sigma = 0.05$, we obtain $\epsilon = 0.47$, while the standard setting achieves $\epsilon = 0.64$ leading to a 1.6× improvement in privacy guarantee.

Figure 2: Improving privacy with $\alpha$.

## 6.3 Benchmarking DP-RSVRG

In this section, we compare our proposed DP-SVRG with DP-RGD and DP-RSGD (Han et al., 2022a) for the task of computing the Fréchet mean and leading eigenvector with privacy configuration $\epsilon = \{0.1, 0.3, 0.5\}$ and $\delta = 10^{-6}$. The parameter details for all the algorithms are in Section C.

**Private Fréchet mean on SPD manifold.** We consider the problem of privately estimating the Fréchet mean of SPD matrices under the Affine-Invariant metric. We select images from PATHMNIST medical imaging dataset (Yang et al., 2021) and pass them through the covariance descriptor pipeline to generate images, each represented as a SPD matrix of size $11 \times 11$. Please refer to Section C.1 for more details on the problem formulation and covariance descriptors. We consider the two sets consisting of 10704 and 10356 images from two different classes. For each set, we compute the optimal Fréchet mean by running the (non-private) RGD for 1000 epochs with learning rate set to 0.5. For both the sets, we plot excess risk against the IFO calls in Figure 3a averaged over five randomized runs. The plots corresponding to the two sets are shown in the two rows of Figure 3a.

**Private principal eigenvector computation on sphere.** We also consider the problem of computing the leading eigenvector a symmetric matrix, details in Section C.2. We take images from two classes of MNIST and generate 784 vectors to form two sets of 6903 and 7877 images. For each set, we compute the covariance matrix and compute its leading eigenvector by using eigen-decomposition of matrix $1/n \sum_{i=1}^n \mathbf{z}_i\mathbf{z}_i^T$ to find the optimal solution. We plot the excess risk against the IFO calls in Figure 3b averaged over five randomized runs. The plots corresponding to the two sets are shown in the two rows of Figure 3b.

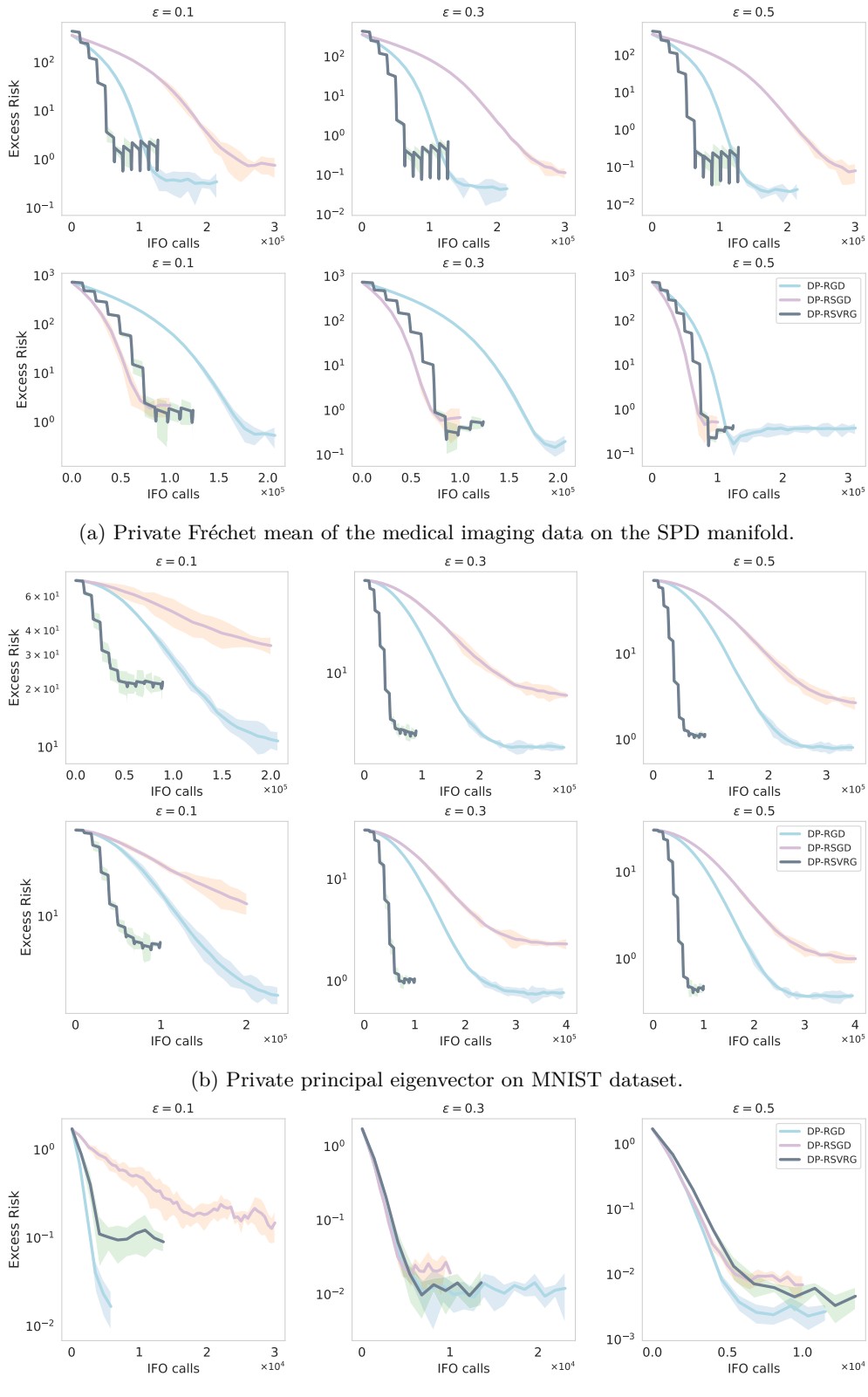

(a) Private Fréchet mean of the medical imaging data on the SPD manifold.

(b) Private principal eigenvector on MNIST dataset.

(c) Private Fréchet mean of Poincaré embeddings.

Figure 3: Comparison between DP-RGD, DP-RSGD, and DP-RSVRG. Each row in (a), (b), and (c) corresponds to consistent dataset. We see the proposed DP-SVRG achieves a comparable excess risk compared to the baselines with lower number of IFO calls.

**Private Fréchet mean of the Poincaré word embeddings.** We generate the hierarchy tree of transitive closure of mammal subtree of the WordNet dataset (Miller, 1995), a lexical database, and compute the private Fréchet mean of the Poincaré word embeddings (Nickel & Kiela, 2017). WordNet provides relationship between pairs of concepts. For instance, the 'mammal' subtree of WordNet has the concept 'mammal' as the root node and the 'is-a' (hypernymy) relationship defines its edges: 'tiger' is-a 'mammal', 'lion' is-a 'rodent', etc. The mammal subtree consists of 1180 nodes and 6540 edges. The results are shown in Figure 3c.

**Results.** In Figure 3a, we observe that the proposed DP-RSVRG obtains an overall better excess risk in computing the private Fréchet mean of the two classes (corresponding to the two rows in Figure 3a) on medical imaging data. In Figure 3a) first row, DP-RSVRG performs consistently better than both the baselines. In Figure 3a) second row, DP-RSVRG performs better than DP-RGD and is similar to DP-SRGD. In Figure 3b, the benefit of variance reduction is clearly observed. In both rows of Figure 3b, the proposed DP-SVRG consistently outperforms DP-RGD and DP-RSGD in the gradient calls and achieves a good excess risk. On the Fréchet mean computation of the Poincaré embeddings (Figure 3c), we see that the benefit of variance reduction as well as the proposed DP-RSVRG performs better than DP-RSGD especially in low $\epsilon$ regime (more stringent private setting). In all cases, DP-RGD performs the best and our proposed DP-RSVRG matches the performance in the initial iterations.

Overall, we observe that the proposed DP-RSVRG obtains better or comparable excess risk against DP-GD and DP-SGD with generally fewer IFO calls across different levels of noise injection.

## 7 Conclusion

In this work, we have improved the framework of differentially private Riemannian optimization via efficient sampling and variance reduction. We have proposed a linear isometry based sampling strategy to generate tangent Gaussian samples. This largely reduces the cost of privatizing Riemannian optimization. In addition, we have shown how variance reduction improves the gradient complexity in practice. We believe this work allows Riemannian optimization to be privatized efficiently for large-scale applications.

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

# A Details about parallel transport and vector transport

## A.1 Parallel transport expressions for SPD, hyperbolic, and sphere manifolds

**SPD manifold.** For the Affine-Invariant and the Log-Euclidean metrics, the parallel transport operation of a tangent vector $\mathbf{U} \in \mathrm{SYM}(m)$ from $\widehat{\mathbf{W}}$ to $\mathbf{W}$, $\widehat{\mathbf{W}}, \mathbf{W} \in \mathrm{SPD}(m)$ is available is closed form (Bhatia, 2009; Pennec et al., 2006; Thanwerdas & Pennec, 2023).

For the Bures-Wasserstein metric, there is no closed-form expression for the parallel transport operation for general $\widehat{\mathbf{W}}, \mathbf{W}$. However, when $\widehat{\mathbf{W}}$ and $\mathbf{W}$ commute, there exists a closed-form expression (Thanwerdas & Pennec, 2023). We exploit this for our case as $\widehat{\mathbf{W}} = \mathbf{I}$ (Algorithm 4), i.e., any base point always commutes with the reference point. Below, we list the parallel transport expressions for all the three metrics, i.e.,

$$\text{Affine-Invariant: } \mathrm{PT}^{\widehat{\mathbf{W}} \to \mathbf{W}}(\mathbf{U}) = (\mathbf{W}\widehat{\mathbf{W}}^{-1})^{\frac{1}{2}} \mathbf{U} (\widehat{\mathbf{W}}^{-1}\mathbf{W})^{\frac{1}{2}},$$

$$\text{Bures-Wasserstein: } \mathrm{PT}^{\widehat{\mathbf{W}} \to \mathbf{W}}(\mathbf{U}) = \mathbf{P} \left[ \mathbf{K}_{\mathrm{BW}} \odot \left( \mathbf{P}^T \mathbf{U} \mathbf{P} \right) \right] \mathbf{P}^T,$$

$$\text{Log-Euclidean: } \mathrm{PT}^{\widehat{\mathbf{W}} \to \mathbf{W}}(\mathbf{U}) = \mathbf{P} \left[ \mathbf{K}_{\mathrm{LE}} \odot \left( \mathbf{P}^T \mathbf{U} \mathbf{P} \right) \right] \mathbf{P}^T,$$

where $\mathbf{K}_{\mathrm{BW}} \in \mathbb{R}^{m \times m}$ such that $(\mathbf{K}_{\mathrm{BW}})_{rs} = \frac{\lambda_r + \lambda_s}{\delta_r + \delta_s}$, $\mathbf{K}_{\mathrm{LE}} \in \mathbb{R}^{m \times m}$ such that $(\mathbf{K}_{\mathrm{LE}})_{rs} = f(\lambda_r, \lambda_s)$. Here, $f(x,y) = \frac{x-y}{\exp(x) - \exp(y)}$ if $x \neq y$ else $f(x,y) = \frac{1}{\exp(x)}$ when $x = y$ and $(\delta_1, \ldots, \delta_m)$ and $(\lambda_1, \ldots, \lambda_m) \in \mathbb{R}^m$ denotes the eigenvalues of $\widehat{\mathbf{W}}$ and $\mathbf{W}$, respectively.

**Hyperbolic manifold.** The parallel transport expressions can be found in (Lou et al., 2020),

$$\text{Poincaré ball: } \mathrm{PT}^{\widehat{\mathbf{w}} \to \mathbf{w}}(\mathbf{u}) = \frac{1 - \|\mathbf{w}\|_2^2}{1 - \|\widehat{\mathbf{w}}\|_2^2} \mathrm{gyr}[\mathbf{w}, -\widehat{\mathbf{w}}](\mathbf{u}), \mathrm{gyr}[\widehat{\mathbf{w}}, \mathbf{w}](\mathbf{u}) = (\mathbf{o} \ominus (\widehat{\mathbf{w}} \oplus \mathbf{w})) \oplus (\widehat{\mathbf{w}} \oplus (\mathbf{w} \oplus \mathbf{u})),$$

$$\text{where } \widehat{\mathbf{w}} \oplus \mathbf{w} = \frac{[(1 + 2\langle\widehat{\mathbf{w}}, \mathbf{w}\rangle_2 + \|\mathbf{w}\|_2^2)\widehat{\mathbf{w}} + (1 - \|\widehat{\mathbf{w}}\|_2^2)\mathbf{w}]}{[1 + 2\langle\widehat{\mathbf{w}}, \mathbf{w}\rangle_2 + \|\widehat{\mathbf{w}}\|_2^2 \|\mathbf{w}\|_2^2]}, \widehat{\mathbf{w}} \ominus \mathbf{w} = \widehat{\mathbf{w}} \oplus -\mathbf{w}.$$

$$\text{Lorentz hyperboloid: } \mathrm{PT}^{\widehat{\mathbf{w}} \to \mathbf{w}}(\mathbf{u}) = \mathbf{u} - \frac{\langle\mathbf{w}, \mathbf{u}\rangle_{\mathcal{L}}}{1 - \langle\widehat{\mathbf{w}}, \mathbf{w}\rangle_{\mathcal{L}}}(\widehat{\mathbf{w}} + \mathbf{w}).$$

**Sphere manifold.** The parallel transport expression can be found in (Absil et al., 2009; Boumal, 2022),

$$\mathrm{PT}^{\widehat{\mathbf{w}} \to \mathbf{w}}(\mathbf{u}) = \left( \mathbf{I} + (\cos\|\mathbf{v}\|_2 - 1)\frac{\mathbf{v}\mathbf{v}^T}{\|\mathbf{v}\|_2} - \sin\|\mathbf{v}\|_2 \frac{\widehat{\mathbf{w}}\mathbf{v}^T}{\|\mathbf{v}\|_2} \right) \mathbf{u},$$

$$\text{where } \mathbf{v} = \mathrm{Exp}_{\widehat{\mathbf{w}}}^{-1}\mathbf{w} = \arccos\langle\widehat{\mathbf{w}}, \mathbf{w}\rangle_2 \frac{(\mathbf{I} - \widehat{\mathbf{w}}\widehat{\mathbf{w}}^T)(\mathbf{w} - \widehat{\mathbf{w}})}{\|(\mathbf{I} - \widehat{\mathbf{w}}\widehat{\mathbf{w}}^T)(\mathbf{w} - \widehat{\mathbf{w}})\|_2}.$$

## A.2 Vector transport for Stiefel and Grassmann manifolds

Efficient vector transport on the Stiefel and Grassmann manifolds are provided in (Huang et al., 2017) which proposes a strategy called transportation by parallelization (Huang et al., 2015). For the exact algorithms, see (Huang et al., 2017, Algorithms 3, 4, and 5).

# B Proofs

## B.1 Proof of Lemma 1

**Theorem 9** (Change of variable formula). *Let $X, Y$ be measurable space and $\phi : X \to Y$ and $f : Y \to \mathbb{R}$ is measurable mapping and let $\lambda$ be measure on $X$ and $\phi_* \lambda$ denote the pushforward measure of $\lambda$ through $\phi$ on $Y$ then $\int_Y f d(\phi_* \lambda) = \int_X f \circ \phi \, d\lambda$.*

*Proof.*
**1.** Let $\vec{\mu} \in \mathbb{R}^d$ denote the coordinates of $\mu$ and consider the normalizing constant

$$
C_{w,\sigma} = \int_{T_w\mathcal{M}} \exp\left(-\frac{\|\nu - \mu\|_w^2}{2\sigma^2}\right) d(\phi_*\lambda)(\nu) \overset{(*)}{=} \int_{\mathbb{R}^d} \exp\left(-\frac{\left\|\sum_{i=1}^d c_i\beta_i - \sum_{i=1}^d \vec{\mu}_i\beta_i\right\|_w^2}{2\sigma^2}\right) d\lambda(c)
$$

$$
= \int_{\mathbb{R}^d} \exp\left(-\frac{\sum_{i=1}^d \sum_{j=1}^d \langle(c_i - \vec{\mu}_i)\beta_i, (c_j - \vec{\mu}_j)\beta_j\rangle_w}{2\sigma^2}\right) d\lambda(c) \overset{(**)}{=} \int_{\mathbb{R}^d} \exp\left(-\frac{\sum_{i=1}^d (c_i - \vec{\mu}_i)^2}{2\sigma^2}\right) d\lambda(c)
$$

$$
\overset{(\dagger)}{=} (2\pi\sigma^2)^{d/2}, \tag{4}
$$

where we use the change of variable rule (Theorem 9) under transformation $\phi$ in $(*)$, that $(\beta_1, \ldots, \beta_d)$ is orthonormal tangent vectors in $(**)$, and that $\int_{\mathbb{R}^d} \exp(-\frac{\sum_{i=1}^d (c_i - \vec{\mu}_i)^2}{2\sigma^2}) d\lambda$ is the normalizing constant of $\mathcal{N}(\vec{\mu}_i, \sigma^2.I)$ in $(\dagger)$.

**2.**

Now, let $\xi \sim \mathcal{N}_w(\mu, \sigma^2)$, we show that $\vec{\xi} \sim \mathcal{N}(\vec{\mu}, \sigma^2 I_d)$. Let $A \subseteq \mathbb{R}^d$ be a measurable set, then consider

$$
\Pr[\vec{\xi} \in A] = \Pr[\xi \in \phi_{\mathscr{B}}(A)] = \int_{\phi_{\mathscr{B}}(A)} \frac{1}{(2\pi\sigma^2)^{d/2}} \exp\left(-\frac{\|\nu - \mu\|_w^2}{2\sigma^2}\right) d(\phi_*\lambda)(\nu)
$$

$$
= \int_A \frac{1}{(2\pi\sigma^2)^{d/2}} \exp\left(-\frac{\sum_{i=1}^d (c_i - \vec{\mu}_i)^2}{2\sigma^2}\right) d\lambda(c).
$$

The last equality is obtained similarly as in (4). Since the last expression is exactly probability that a random vector distributed as $\mathcal{N}(\vec{\mu}, \sigma^2 I_d)$ belongs to set $A$, we are done. The converse is shown in a similar way.

**3.** This simply follows Statement **2** of Lemma 1 and the variance bound from the standard Gaussian distribution.

$\square$

## B.2 Proof of Claim 2

*Proof.* Given that $\text{LI}^{\widehat{w}\to w}$ is a linear isometric mapping, one can show that it is invertible and its inverse is again isometry, which we will denote by $\text{LI}^{w\to\widehat{w}}$. If $\phi_*\lambda$ is Lebesuge measure on $T_{w_1}\mathcal{M}$ then $\text{LI}^{w_1\to w}{}_*(\phi_*\lambda)$ is the Lebesgue measure on $T_w\mathcal{M}$. This can be seen by observation that, if $\mathscr{B} = \{\beta_1, \ldots, \beta_d\}$ is orthonormal basis for $T_{\widehat{w}}\mathcal{M}$ then $\{\text{LI}^{\widehat{w}\to w}\beta_1, \ldots, \text{LI}^{\widehat{w}\to w}\beta_d\}$ is orthonormal basis for $T_w\mathcal{M}$. Let $\xi \sim \mathcal{N}_{\widehat{w}}(\mu, \sigma^2)$, we will show that $\text{LI}^{\widehat{w}\to w}\xi \sim \mathcal{N}_w(\text{LI}^{\widehat{w}\to w}\mu, \sigma^2)$. consider measurable set $S \subseteq T_w\mathcal{M}$

$$
\Pr\left[\text{LI}^{\widehat{w}\to w}(\xi_1) \in S\right] = \Pr\left[\xi_1 \in \text{LI}^{w\to\widehat{w}}(S)\right] = \int_{\text{LI}^{w\to\widehat{w}}(S)} \frac{1}{(2\pi\sigma^2)^{d/2}} \exp\left(-\frac{\|\nu - \mu\|_{\widehat{w}}^2}{2\sigma^2}\right) d(\phi_*\lambda)(\nu)
$$

$$
\overset{(*)}{=} \int_S \frac{1}{(2\pi\sigma^2)^{d/2}} \exp\left(-\frac{\left\|\text{LI}^{w\to\widehat{w}}(\nu) - \mu)\right\|_{\widehat{w}}^2}{2\sigma^2}\right) d\left(\text{LI}^{\widehat{w}\to w}_*(\phi_*\lambda)\right)(\nu)
$$

$$
\overset{(**)}{=} \int_S \frac{1}{(2\pi\sigma^2)^{d/2}} \exp\left(-\frac{\left\|\nu - \text{LI}^{\widehat{w}\to w}(\mu)\right\|_w^2}{2\sigma^2}\right) d\left(\text{LI}^{\widehat{w}\to w}_*(\phi_*\lambda)\right)(\nu),
$$

where we used change of variables formula Theorem 9 (with $X = \text{LI}^{w\to\widehat{w}}(S), Y = S$ and $\phi = \text{LI}^{\widehat{w}\to w}$) and that LI is isometry in $(**)$. Since $\text{LI}^{\widehat{w}\to w}{}_*(\phi_*\lambda)$ is the Lebesuge measure on $T_w\mathcal{M}$, we have that $\text{LI}^{\widehat{w}\to w}\xi \sim \mathcal{N}_w(\text{LI}^{\widehat{w}\to w}\mu, \sigma^2)$.

$\square$

## B.3 Proofs of Section 5

### B.3.1 Proof of Claim 3

*Proof.* Let $\mathcal{Q}^{s+1}$ denote the full gradient query given by $\mathcal{Q}^{s+1}(Z) = \frac{1}{n}\sum_{i=1}^{n}\operatorname{grad} f(\widetilde{w}^s; z_i)$. Let $Z, Z' \in \mathcal{Z}^n$ denote adjacent datasets, consider the sensitivity, denoted as $\Delta^s$,

$$\Delta^{s+1} = \sup_{Z \sim Z'} \|\mathcal{Q}^{s+1}(Z) - \mathcal{Q}^{s+1}(Z')\| \le \frac{1}{n}\left[\|\operatorname{grad} f(\widetilde{w}^s; z_n)\|_{\widetilde{w}^s} + \|\operatorname{grad} f(\widetilde{w}^s; z_n')\|_{\widetilde{w}^s}\right] \le \frac{2\mathcal{C}_0}{n}. \quad (5)$$

Following (Han et al., 2022a, Lemma 2), the moments bound of the full gradient mechanism $\mathcal{R}^s$ is given by

$$\mathcal{K}_{\mathcal{R}^s}(\lambda) \le \frac{\lambda(\lambda+1)}{2\sigma_1^2}(\Delta^s)^2 \overset{(5)}{\le} \frac{2\lambda(\lambda+1)\mathcal{C}_0^2}{n^2\sigma_1^2}.$$

Let $\mathcal{Q}_t^{s+1}$ denote the variance reduced stochastic gradient query given by $\mathcal{Q}_t^{s+1}(Z) = \operatorname{grad} f(w_t^{s+1}; z) - \operatorname{PT}^{\widetilde{w}^s \to w_t^{s+1}}(\operatorname{grad} f(\widetilde{w}^s; z) - g^{s+1})$. Let $Z, Z' \in \mathcal{Z}$ denote adjacent datasets, consider its sensitivity, denoted at $\Delta_{t+1}^s$,

$$\Delta_t^{s+1}$$
$$= \sup_{Z \sim Z'} \left\|\mathcal{Q}_{t2}^{s+1}(Z) - \mathcal{Q}_{t2}^{s+1}(Z')\right\|_{w_t^{s+1}}$$
$$\overset{(*)}{\le} \sup_{Z \sim Z'} \left[\left\|\operatorname{grad} f(w_t^{s+1}; z) - \operatorname{grad} f(w_t^{s+1}; z')\right\|_{w_t^{s+1}} + \left\|\operatorname{PT}^{\widetilde{w}^s \to w_t^{s+1}}(\operatorname{grad} f(\widetilde{w}^s; z) - \operatorname{grad} f(\widetilde{w}^s; z'))\right\|_{w_t^{s+1}}\right]$$
$$\overset{(\dagger)}{=} \sup_{Z \sim Z'} \left[\left\|\operatorname{grad} f(w_t^{s+1}; z) - \operatorname{grad} f(w_t^{s+1}; z')\right\|_{w_t^{s+1}} + \left\|\operatorname{grad} f(\widetilde{w}^s; z) - \operatorname{grad} f(\widetilde{w}^s; z')\right\|_{\widetilde{w}^s}\right]$$
$$\le \sup_{Z \sim Z'} \left[\left\|\operatorname{grad} f(w_t^{s+1}; z)\right\|_{w_t^{s+1}} + \left\|\operatorname{grad} f(w_t^{s+1}; z')\right\|_{w_t^{s+1}} + \left\|\operatorname{grad} f(\widetilde{w}^s; z)\right\|_{\widetilde{w}^s} + \left\|\operatorname{grad} f(\widetilde{w}^s; z')\right\|_{\widetilde{w}^s}\right]$$
$$\overset{(\ddagger)}{\le} 4\mathcal{C}_1, \quad (6)$$

where we used linearity of parallel transport and triangle's inequality in $(*)$ and that parallel transport is isometric in $(\dagger)$ and triangle inequality and assumption of lipschitz in $(\ddagger)$. Now moments bound of $\mathcal{R}_t^{s+1}$ is given by,

$$\mathcal{K}_{\mathcal{R}_t^{s+1}}(\lambda) \le \frac{\lambda(\lambda+1)}{2\sigma_2^2}(\Delta_t^{s+1})^2 \overset{(6)}{\le} \frac{8\lambda(\lambda+1)\mathcal{C}_1^2}{\sigma_2^2}. \quad (7)$$

$\square$

### B.3.2 Proof of Claim 4

*Proof.* By using (Wang et al., 2019b, Lemma 3.7) and by choice of parameters $\sigma^2, \lambda$ we have

$$\mathcal{K}_{\mathrm{sub}\,\mathcal{R}_t^{s+1}}(\lambda) \le \frac{3.5}{n^2}\mathcal{K}_{\mathcal{R}_t^{s+1}}(\lambda) \overset{(7)}{\le} \frac{28\lambda(\lambda+1)\mathcal{C}_1^2}{\sigma^2 n^2}.$$

$\square$

### B.3.3 Proof of Claim 5

*Proof.* For $\mathcal{R}$ can be show $(\epsilon, \delta)$-differentially private by solving for $\epsilon$ and $\delta$ as follows, i.e.,

$$\min_{\alpha \in (0,1)} \frac{mS\lambda(\lambda+1)\mathcal{C}^2}{n^2\sigma^2}\left[\frac{2}{\alpha} + \frac{28}{1-\alpha}\right] = \frac{mS\lambda(\lambda+1)\mathcal{C}^2}{n^2\sigma^2}\left[\frac{2}{\alpha^*} + \frac{28}{1-\alpha^*}\right] \le \frac{\lambda\epsilon}{2}, \exp\left(-\frac{\lambda\epsilon}{2}\right) \le \delta, \quad (8)$$

where $\alpha^* = (\sqrt{14} - 1)/13$ and there exists constant $c_1 > 0$ such that $\sigma^2 \geq c_1 \frac{mS \log(1/\delta)\mathcal{C}^2}{n^2\epsilon^2}$ satisfies (8). Hence, Algorithm 11 satisfies $(\epsilon, \delta)$-DP. For Algorithm 12 using similar arguments there exists constant $c_2 > 0$ such that $\sigma^2 \geq c_2 \frac{mSK \log(1/\delta)\mathcal{C}^2}{n^2\epsilon^2}$ guarantees $(\epsilon, \delta)$-DP . $\qquad\square$

## B.4   Proofs of Section 5.2

**Lemma 10** (Trigonometric distance bound (Zhang & Sra, 2016)). *Let $w_0, w_1, w_2 \in \mathcal{W} \subseteq \mathcal{M}$ lie in totally normal neighborhood of Riemannian manifold with curvature lower bounded by $\kappa_{min}$ and $\ell_0 = \text{dist}(w_0, w_1)$ and $\ell_1 = \text{dist}(w_1, w_2)$ and $\ell_2 = \text{dist}(w_0, w_2)$. Denote $\theta$ as the angle on $T_{w_0}\mathcal{M}$ such that $\cos(\theta) = \frac{1}{\ell_0 \ell_1}\langle \text{Exp}_{w_0}^{-1}(w_1), \text{Exp}_{w_0}^{-1}(w_2)\rangle_{w_0}$. Let $D_{\mathcal{W}}$ be the diameter of $\mathcal{W}$ i.e., $D_{\mathcal{W}} := \max_{w, w'} \text{dist}(w, w')$. Define curvature constant $\zeta = \frac{\sqrt{\kappa_{\min}}}{\tanh\sqrt{\kappa_{\min}}}$ if $\kappa_{\min} < 0$ and $\zeta = 1$ if $\kappa_{\min} \geq 0$. Then, we have that $\ell_1^2 \leq \zeta\ell_0^2 + \ell_2^2 - 2\ell_0\ell_2\cos\theta$.*

**Lemma 11.**

$$\mathbb{E}_{i_t, \epsilon_t}\|v_t^{s+1}\|_{w_t^{s+1}}^2 \leq \mathbb{E}_{i_t}\|\text{grad} f(w_t^{s+1}; z_{i_t}) - \text{PT}^{\widetilde{w}^s \to w_t^{s+1}}(\text{grad} f(\widetilde{w}^s; z_{i_t}) - g^{s+1})\|_{w_t^{s+1}}^2 + d\sigma^2. \quad (9)$$

*Proof.*

$$
\begin{aligned}
\mathbb{E}_{i_t, \epsilon_t}\|v_t^{s+1}\|_{w_t^{s+1}}^2 &= \mathbb{E}_{i_t, \epsilon_t}\|\text{grad} f(w_t^{s+1}; z_{i_t}) - \text{PT}^{\widetilde{w}^s \to w_t^{s+1}}(\text{grad} f(\widetilde{w}^s; z_{i_t}) - g^{s+1}) + \epsilon_t\|_{w_t^{s+1}}^2 \\
&= \mathbb{E}_{i_t, \epsilon_t}\|\text{grad} f(w_t^{s+1}; z_{i_t}) - \text{PT}^{\widetilde{w}^s \to w_t^{s+1}}(\text{grad} f(\widetilde{w}^s; z_{i_t}) - g^{s+1})\|_{w_t^{s+1}}^2 + \mathbb{E}_{\epsilon_t}\|\epsilon_t\|_{w_t^{s+1}}^2 \\
&\quad + \langle \mathbb{E}_{i_t}\text{grad} f(w_t^{s+1}; z_{i_t}) - \text{PT}^{\widetilde{w}^s \to w_t^{s+1}}(\text{grad} f(\widetilde{w}^s; z_{i_t}) - g^{s+1}), \mathbb{E}_{\epsilon_t}[\epsilon_t]\rangle_{w_t^{s+1}} \\
&\leq \mathbb{E}_{i_t}\|\text{grad} f(w_t^{s+1}; z_{i_t}) - \text{PT}^{\widetilde{w}^s \to w_t^{s+1}}(\text{grad} f(\widetilde{w}^s; z_{i_t}) - g^{s+1})\|_{w_t^{s+1}}^2 + d\sigma^2,
\end{aligned}
$$

where we used that $\mathbb{E}_{\epsilon_t}[\epsilon_t] = 0$ and $\mathbb{E}_{\epsilon_t}\|\epsilon_t\|_{w_t^{s+1}}^2 \leq d\sigma^2$ in last inequality. $\qquad\square$

### B.4.1   Proof of Theorem 6

*Proof.* We bound first term $\mathbb{E}_{i_t}\|\text{grad} f(w_t^{s+1}; z_{i_t}) - \text{PT}^{\widetilde{w}^s \to w_t^{s+1}}(\text{grad} f(\widetilde{w}^s; z_{i_t}) - g^{s+1})\|_{w_t^{s+1}}^2$ as in (Zhang et al., 2016)

$$
\begin{aligned}
&\mathbb{E}_{i_t}\left\|\text{grad} f(w_t^{s+1}; z_{i_t}) - \text{PT}^{\widetilde{w}^s \to w_t^{s+1}}(\text{grad} f(\widetilde{w}^s; z_{i_t}) - g^{s+1})\right\|_{w_t^{s+1}}^2 \\
&\leq \mathbb{E}_{i_t}\left\|\text{grad} f(w_t^{s+1}; z_{i_t}) - \text{PT}^{\widetilde{w}^s \to w_t^{s+1}}\text{grad} f(\widetilde{w}^s; z_{i_t}) + \text{PT}^{\widetilde{w}^s \to w_t^{s+1}}\left(\text{grad} F(\widetilde{w}^s) - \text{PT}^{\widetilde{w}^* \to \widetilde{w}^s}\text{grad} F(w^*)\right)\right\|_{w_t^{s+1}}^2 \\
&\leq 2\mathbb{E}_{i_t}\left\|\text{grad} f(w_t^{s+1}; z_{i_t}) - \text{PT}^{\widetilde{w}^s \to w_t^{s+1}}\text{grad} f(\widetilde{w}^s; z_{i_t})\right\|_{w_t^{s+1}}^2 \\
&\quad + 2\mathbb{E}_{i_t}\left\|\text{PT}^{\widetilde{w}^s \to w_t^{s+1}}\left(\text{grad} F(\widetilde{w}^s) - \text{PT}^{\widetilde{w}^* \to \widetilde{w}^s}\text{grad} F(w^*)\right)\right\|_{w_t^{s+1}}^2 \\
&= 2\mathbb{E}_{i_t}\left\|\text{grad} f(w_t^{s+1}; z_{i_t}) - \text{PT}^{\widetilde{w}^s \to w_t^{s+1}}\text{grad} f(\widetilde{w}^s; z_{i_t})\right\|_{w_t^{s+1}}^2 + 2\mathbb{E}_{i_t}\left\|\text{grad} F(\widetilde{w}^s) - \text{PT}^{\widetilde{w}^* \to \widetilde{w}^s}\text{grad} F(w^*)\right\|_{\widetilde{w}^s}^2 \\
&\leq 4L^2\|\text{Exp}_{w_t^{s+1}}^{-1}(w^*)\|_{w_t^{s+1}}^2 + 6L^2\left\|\text{Exp}_{\widetilde{w}^s}^{-1}w^*\right\|_{\widetilde{w}^s}^2 \\
&= 4L^2\text{dist}^2(w_t^{s+1}, w^*) + 6L^2\text{dist}^2(\widetilde{w}^s, w^*). \quad (10)
\end{aligned}
$$

Using the trigonometric distance bound in Lemma 10 with $w_0 = x_t^{s+1}, w_1 = w_{t+1}^{s+1}, w_2 = w^*$,

$$
\begin{aligned}
\text{dist}^2(w_{t+1}^{s+1}, w^*) &\leq \zeta\text{dist}^2(w_{t+1}^{s+1}, w_t^{s+1}) + \text{dist}^2(w_t^{s+1}, w^*) - 2\langle \text{Exp}_{x_t^{s+1}}^{-1}(w_{t+1}^{s+1}), \text{Exp}_{w_t^{s+1}}^{-1}(w^*)\rangle_{w_t^{s+1}} \\
&= \zeta\left\|\text{Exp}_{w_t^{s+1}}^{-1}w_{t+1}^{s+1}\right\|_{w_t^{s+1}}^2 + \text{dist}^2(w_t^{s+1}, w^*) - 2\langle -\eta v_t^{s+1}, \text{Exp}_{w_t^{s+1}}^{-1}(w^*)\rangle_{w_t^{s+1}} \\
&= \zeta\eta^2\left\|v_t^{s+1}\right\|_{w_t^{s+1}}^2 + \text{dist}^2(w_t^{s+1}, w^*) + 2\eta\langle v_t^{s+1}, \text{Exp}_{w_t^{s+1}}^{-1}(w^*)\rangle_{w_t^{s+1}}.
\end{aligned}
$$

Applying expectation we have

$$
\begin{aligned}
&\mathrm{dist}^2(w_{t+1}^{s+1}, w^*)\\
&\leq \zeta\eta^2 \mathbb{E}_{i_t,\epsilon_t}\left\|v_t^{s+1}\right\|^2_{w_t^{s+1}} + \mathrm{dist}^2(w_t^{s+1}, w^*) + 2\eta\langle \mathbb{E}_{i_t,\epsilon_t} v_t^{s+1}, \mathrm{Exp}^{-1}_{w_t^{s+1}}(w^*)\rangle_{w_t^{s+1}}\\
&= \zeta\eta^2 L^2\left[4\mathrm{dist}^2(w_t^{s+1}, w^*) + 6\mathrm{dist}^2(\widetilde{w}^s, w^*)\right] + 2\eta\langle \mathrm{grad}\, F(w_t^{s+1}), \mathrm{Exp}^{-1}_{w_t^{s+1}}(w^*)\rangle_{w_t^{s+1}} + d\zeta\eta^2\sigma^2\\
&\leq \zeta\eta^2 L^2\left[4\mathrm{dist}^2(w_t^{s+1}, w^*) + 6\mathrm{dist}^2(\widetilde{w}^s, w^*)\right] + 2\eta[F(w^*) - F(w_t^{s+1}) - \frac{\mu}{2}\mathrm{dist}^2(w_t^{s+1}, w^*)] + d\zeta\eta^2\sigma^2\\
&\leq (1 + 4\zeta\eta^2 L^2 - \eta\mu)\mathrm{dist}^2(w_t^{s+1}, w^*) + 6\zeta\eta^2 L^2\mathrm{dist}^2(\widetilde{w}^s, w^*) + d\zeta\eta^2\sigma^2.
\end{aligned}
$$

Defining $u_t = \mathrm{dist}^2(w_{t+1}^{s+1}, w^*)$, $q = (1 + 4\zeta\eta^2 L^2 - \eta\mu)$, $p = 6\zeta\eta^2 L^2$, $c = d\zeta\eta^2\sigma^2$ we have following recurrence $u_{t+1} - p u_0 \leq q(u_t - p u_0) + c$ from which we have that $u_m \leq (p + q^m(1-p))u_0 + \sum_{i=1}^{m-1} q^i c$. Now choosing $\eta = \frac{\mu}{17\zeta L^2}$ and $m \geq \frac{10\zeta L^2}{\mu^2}$. we get $q = 1 - \frac{\mu^2}{10\zeta L^2}$ and $p = 1/5$. Note that $0 < \frac{\mu^2}{10\zeta L^2} < 1$ ( $L > \mu$, $\zeta \geq 1$) and hence $0 < q < 1$ and from which we have that $(p + q^m(1-p)) = 1/2$.

$$
\begin{aligned}
\mathbb{E}[d^2(w_m^{s+1}, w^*)] &\leq \mathbb{E}[\mathrm{dist}^2(w_m^s, w^*)] + d\zeta\frac{\mu^2\sigma^2}{289\zeta^2 L^4}\sum_{i=1}^{m-1}\left(1 - \frac{\mu^2}{10\zeta L^2}\right)^i\\
&\leq \mathbb{E}[\mathrm{dist}^2(w_m^s, w^*)] + d\zeta\frac{\mu^2\sigma^2}{289\zeta^2 L^4}\sum_{i=1}^{\infty}\left(1 - \frac{\mu^2}{10\zeta L^2}\right)^i\\
&= \mathbb{E}[\mathrm{dist}^2(w_m^s, w^*)] + d\zeta\frac{\mu^2\sigma^2}{289\zeta^2 L^4}\frac{10\zeta L^2}{\mu^2} = \mathbb{E}[\mathrm{dist}^2(w_m^s, w^*)] + d\frac{10\sigma^2}{289 L^2},
\end{aligned}
$$

from which we have

$$
\begin{aligned}
\mathbb{E}[\mathrm{dist}^2(w_m^S, w^*)] &= 2^{-S}\mathbb{E}[\mathrm{dist}^2(w_m^0, w^*)] + d\frac{10\sigma^2}{289 L^2}\sum_{i=0}^{S}\frac{1}{2^i} \leq 2^{-S}\mathbb{E}[\mathrm{dist}^2(w_m^0, w^*)] + 2dc^{-1}\frac{10}{289 L^2}\frac{mS\log(1/\delta)L_0^2}{n^2\epsilon^2}\\
&\leq 2^{-S}\mathbb{E}[\mathrm{dist}^2(w_m^0, w^*)] + d\frac{200\zeta}{289\mu^2}\frac{S\log(1/\delta)L_0^2}{n^2\epsilon^2}.
\end{aligned}
$$

$$
\mathbb{E}\left[f(x^a) - f(w^*)\right] \leq \frac{1}{2}\mathbb{E}\left[L\mathrm{dist}^2(x_a, w^*)\right] \leq 2^{-S}L\mathbb{E}[\mathrm{dist}^2(w^0, w^*)] + Ld\frac{\zeta}{\mu^2}\frac{S\log(1/\delta)L_0^2}{n^2\epsilon^2}.
$$

Now, setting $2^{-S} = d\frac{\zeta}{\mu^2}\frac{\log(1/\delta)L_0^2}{n^2\epsilon^2\mathbb{E}[\mathrm{dist}^2(w^0, w^*)]}$ $\implies$ $2^S = \frac{n^2\epsilon^2 289\mu^2\mathbb{E}[\mathrm{dist}^2(w^0, w^*)]}{d100\zeta\log(1/\delta)L_0^2}$ $\implies$ $S = \mathcal{O}\left(\log\left(\frac{n\epsilon\mu\mathbb{E}[\mathrm{dist}^2(w^0, w^*)]}{\log(1/\delta)\zeta L_0 d}\right)\right)$, substituting this we have that, and now for $S = \mathcal{O}\left(\log\left(\frac{n\epsilon\mu\mathbb{E}[\mathrm{dist}^2(w^0, w^*)]}{\log(1/\delta)\zeta L_0^2 d}\right)\right)$

$$
\mathbb{E}\left[f(x^a) - f(w^*)\right] \leq \mathcal{O}\left(\frac{d\zeta LL_0^2\log(1/\delta)\mathbb{E}[\mathrm{dist}^2(w^0, w^*)]}{\mu^2 n^2\epsilon^2}\log\left(\frac{n\epsilon\mu}{\zeta L_0^2 d\log(1/\delta)}\right)\right). \tag{11}
$$

**Gradient complexity:** $S \times n$ plus $m \times 2$ IFO calls $= 2nS + 2mS$,

$$
\mathcal{O}\left(\left(n + \frac{\zeta L^2}{\mu^2}\right)\log\left(\frac{n\epsilon\mu\mathbb{E}[\mathrm{dist}^2(w^0, w^*)]}{\log(1/\delta)\zeta L_0 d}\right)\right). \tag{12}
$$

This completes the proof. $\qquad\square$

### B.4.2 Proof of Theorem 7

Before proving Theorem 7, we state and prove following lemma that we will be using later.

**Lemma 12.** *Assume that each $f_i$ is $L$-g-smooth, the sectional curvature in $\mathcal{X}$ is lower bounded by $\kappa_{min}$ and we run Algorithm 11 with Option II. For $c_t, c_{t+1}, \beta, \eta > 0$ and suppose we have $c_t = c_{t+1}(1 + \beta\eta + 2\zeta L^2\eta^2) + L^3\eta^2$ and $\delta(t) = \eta - \frac{c_{t+1}\eta}{\beta} - L\eta^2 - 2c_{t+1}\zeta\eta^2 > 0$, then the iterate $w_t^{s+1}$ satisfies the bound*

$$\mathbb{E}\|\operatorname{grad} f(w_t^{s+1})\|^2 \leq \frac{R_t^{s+1} - R_{t+1}^{s+1}}{\delta_t} + \frac{\left(\frac{1}{2}dL\eta^2 + c_{t+1}\zeta d\eta^2\right)}{\delta_t}\sigma^2,$$

*where $R_t^{s+1} := \mathbb{E}[F(w_t^{s+1}) + c_t \left\|\operatorname{Exp}_{\widetilde{w}^s} w_t^{s+1}\right\|]$ for $0 \leq s \leq S - 1$.*

*Proof.* The proof is adapted from (Zhang et al., 2016, Lemma 2). Denoting $\Delta_t^{s+1} = \operatorname{grad} f(w_t^{s+1}; z_{i_t}) - \operatorname{PT}^{\widetilde{w}^s \to w_t^{s+1}} \operatorname{grad} f(\widetilde{w}^s; z_{i_t})$ it can be seen that $\mathbb{E}_{i_t | \widetilde{x}^s, w_t^{s+1}}[\Delta_t^{s+1}] = \operatorname{grad} F(w_t^{s+1}) - \operatorname{PT}^{\widetilde{w}^s \to w_t^{s+1}} \operatorname{grad} F(\widetilde{w}^s)$

$$\mathbb{E}_{i_t, \epsilon_t} \left\|v_t^{s+1}\right\|_{w_t^{s+1}}^2 \overset{(9)}{\leq} \mathbb{E}_{i_t} \left\|\operatorname{grad} f(w_t^{s+1}; z_{i_t}) - \operatorname{PT}^{\widetilde{w}^s \to w_t^{s+1}}(\operatorname{grad} f(\widetilde{w}^s; z_{i_t}) - g^{s+1})\right\|_{w_t^{s+1}}^2 + d\sigma^2$$

$$= \mathbb{E}_{i_t} \left\|\Delta_t^{s+1} - \mathbb{E}_{i_t}\Delta_t^{s+1} + \operatorname{grad} F(w_t^{s+1})\right\|_{w_t^{s+1}}^2 + d\sigma^2$$

$$\overset{(*)}{\leq} 2\mathbb{E}_{i_t} \left\|\Delta_t^{s+1} - \mathbb{E}_{i_t}\Delta_t^{s+1}\right\|^2 + 2\left\|\operatorname{grad} F(w_t^{s+1})\right\|_{w_t^{s+1}}^2 + d\sigma^2$$

$$\overset{(**)}{\leq} 2\mathbb{E}_{i_t} \left\|\Delta_t^{s+1}\right\|_{w_t^{s+1}}^2 + 2\left\|\operatorname{grad} F(w_t^{s+1})\right\|_{w_t^{s+1}}^2 + d\sigma^2$$

$$\overset{(\dagger)}{\leq} 2L^2 \left\|\operatorname{Exp}_{\widetilde{w}^s}^{-1}(w_t^{s+1})\right\|_{\widetilde{w}^s}^2 + 2\left\|\operatorname{grad} F(w_t^{s+1})\right\|_{w_t^{s+1}}^2 + d\sigma^2,$$

where $\|a + b\|^2 \leq 2\|a\|^2 + 2\|b\|^2$ in $(*)$ and $\mathbb{E}_{i_t} \left\|\Delta_t^{s+1} - \mathbb{E}_{i_t}\Delta_t^{s+1}\right\|^2 = \mathbb{E}_{i_t} \left\|\Delta_t^{s+1}\right\|^2 - \left\|\mathbb{E}\Delta_t^{s+1}\right\|^2 \leq \mathbb{E}_{i_t} \left\|\Delta_t^{s+1}\right\|^2$ in $(**)$ and assumption that $f_i$ is $L$-g-smooth in $(\dagger)$. Taking full expectation we have

$$\mathbb{E} \left\|v_t^{s+1}\right\|_{w_t^{s+1}}^2 \leq 2L^2 \left\|\operatorname{Exp}_{\widetilde{w}^s}^{-1}(w_t^{s+1})\right\|_{\widetilde{w}^s}^2 + 2\left\|\operatorname{grad} F(w_t^{s+1})\right\|_{w_t^{s+1}}^2 + d\sigma^2. \tag{13}$$

For bounding the Lyapunov function $R_{t+1}^{s+1} := \mathbb{E}\left[F(w_{t+1}^{s+1}) + c_{t+1} \left\|\operatorname{Exp}_{\widetilde{w}^s}(w_{t+1}^{s+1})\right\|^2\right]$, we need to bound on $\mathbb{E}[F(w_{t+1}^{s+1})]$, $\mathbb{E}[\left\|\operatorname{Exp}_{\widetilde{w}^s}(w_{t+1}^{s+1})\right\|^2]$, First consider

$$\mathbb{E}\left[F(w_{t+1}^{s+1})\right]$$

$$\overset{(*)}{\leq} \mathbb{E}\left[F(w_t^{s+1}) + \left\langle\operatorname{grad} F(w_t^{s+1}), \operatorname{Exp}_{w_t^{s+1}}^{-1}(w_{t+1}^{s+1})\right\rangle_{w_t^{s+1}} + \frac{L}{2}\left\|\operatorname{Exp}_{w_t^{s+1}}^{-1}(w_{t+1}^{s+1})\right\|_{w_t^{s+1}}^2\right]$$

$$\overset{(**)}{\leq} \mathbb{E}\left[F(w_t^{s+1}) - \eta\left\|\operatorname{grad} F(w_t^{s+1})\right\|_{w_t^{s+1}}^2 + \frac{L\eta^2}{2}\left\|v_t^{s+1}\right\|_{w_t^{s+1}}^2\right]$$

$$\overset{(13)}{\leq} \mathbb{E}\left[F(w_t^{s+1}) - \eta\left\|\operatorname{grad} F(w_t^{s+1})\right\|_{w_t^{s+1}}^2 + \frac{L\eta^2}{2}\left(2L^2\|\operatorname{Exp}_{\widetilde{w}^s}^{-1}(w_t^{s+1})\|^2 + 2\|\operatorname{grad} F(w_t^{s+1})\|^2 + \sigma^2 d\right)\right]$$

$$= (L\eta^2 - \eta)\|\operatorname{grad} F(w_t^{s+1})\|^2 + F(w_t^{s+1}) + L^3\eta^2\|\operatorname{Exp}_{\widetilde{w}^s}^{-1}(w_t^{s+1})\|^2 + \frac{1}{2}dL\eta^2\sigma^2, \tag{14}$$

where we used the assumption that $f_i$ is $L$-g-smooth implies that $F$ is $L$-g-smooth in $(*)$ and $\operatorname{Exp}_{w_t^{s+1}}^{-1} = v_t^{s+1}$ and $\mathbb{E}\left[v_t^{s+1}\right] = \operatorname{grad} F(w_t^{s+1})$ in $(**)$. Using the trigonometric distance bound on $w_t^{s+1}, w_{t+1}^{s+1}, \widetilde{w}^s$ we have,

$$\left\|\operatorname{Exp}_{\widetilde{w}^s}^{-1}(w_{t+1}^{s+1})\right\|_{\widetilde{w}^s}^2 \leq \left\|\operatorname{Exp}_{\widetilde{w}^s}^{-1}(w_t^{s+1})\right\|_{\widetilde{w}^s}^2 + \zeta\left\|\operatorname{Exp}_{w_t^{s+1}}^{-1}(w_{t+1}^{s+1})\right\|_{w_t^{s+1}}^2 - \left\langle\operatorname{Exp}_{w_t^{s+1}}^{-1}(w_{t+1}^{s+1}), \operatorname{Exp}_{w_t^{s+1}}^{-1}(\widetilde{w}^s)\right\rangle_{w_t^{s+1}}$$

$$= \left\|\operatorname{Exp}_{\widetilde{w}^s}^{-1}(w_t^{s+1})\right\|^2 + \zeta\eta^2\left\|v_t^{s+1}\right\|^2 + 2\eta\langle\operatorname{grad} F(w_t^{s+1}), \operatorname{Exp}_{w_t^{s+1}}^{-1}(\widetilde{w}^s)\rangle.$$

Taking the expectation we have

$$
\mathbb{E} \left\| \mathrm{Exp}_{\widetilde{w}^s}^{-1}(w_{t+1}^{s+1}) \right\|_{\widetilde{w}^s}^2
$$
$$
\leq \mathbb{E} \left[ \left\| \mathrm{Exp}_{\widetilde{w}^s}^{-1}(w_t^{s+1}) \right\|^2 + \zeta\eta^2 \|v_t^{s+1}\|^2 + 2\eta\langle \mathrm{grad}\, F(w_t^{s+1}), \mathrm{Exp}_{w_t^{s+1}}^{-1}(\widetilde{w}^s)\rangle \right]
$$
$$
\leq \mathbb{E} \left[ \left\| \mathrm{Exp}_{\widetilde{w}^s}^{-1}(w_t^{s+1}) \right\|^2 + \zeta\eta^2 \left\| v_t^{s+1} \right\|^2 + 2\eta \left[ \frac{1}{2\beta} \left\| \mathrm{grad}\, f(w_t^{s+1}) \right\|^2 + \frac{\beta}{2} \left\| \mathrm{Exp}_{w_t^{s+1}}^{-1}(\widetilde{w}^s) \right\|^2 \right] \right]
$$
$$
\leq \mathbb{E} \left[ (1 + \beta\eta) \left\| \mathrm{Exp}_{\widetilde{w}^s}^{-1}(w_t^{s+1}) \right\|^2 + \zeta\eta^2 \left[ 2L^2 \left\| \mathrm{Exp}_{\widetilde{w}^s}^{-1}(w_t^{s+1}) \right\|^2 + 2 \left\| \mathrm{grad}\, F(w_t^{s+1}) \right\|^2 + \sigma^2 d \right] \right]
$$
$$
+ \mathbb{E} \left[ \frac{\eta}{\beta} \left\| \mathrm{grad}\, f(w_t^{s+1}) \right\|^2 \right]
$$
$$
= \left( 1 + 2\zeta\eta^2 L^2 + \eta\beta \right) \left\| \mathrm{Exp}_{\widetilde{w}^s}^{-1}(w_t^{s+1}) \right\|^2 + \left( 2\zeta\eta^2 + \frac{\eta}{\beta} \right) \left\| \mathrm{grad}\, F(w_t^{s+1}) \right\|^2 + \zeta d\eta^2 \sigma^2. \tag{15}
$$

Putting (14) and (15) into $R_{t+1}^{s+1}$, we have

$$
R_{t+1}^{s+1} := \mathbb{E}[f(w_{t+1}^{s+1}) + c_{t+1} \left\| \mathrm{Exp}_{\widetilde{w}^s}^{-1}(w_{t+1}^{s+1}) \right\|^2]
$$
$$
= c_{t+1} \left( 1 + 2\zeta\eta^2 L^2 + \eta\beta \right) \left\| \mathrm{Exp}_{\widetilde{w}^s}^{-1}(w_t^{s+1}) \right\|^2 + c_{t+1} \left( 2\zeta\eta^2 + \frac{\eta}{\beta} \right) \left\| \mathrm{grad}\, F(w_t^{s+1}) \right\|^2 + c_{t+1}\zeta d\eta^2\sigma^2
$$
$$
+ (L\eta^2 - \eta) \left\| \mathrm{grad}\, F(w_t^{s+1}) \right\|^2 + F(w_t^{s+1}) + L^3\eta^2 \left\| \mathrm{Exp}_{\widetilde{w}^s}^{-1}(w_t^{s+1}) \right\|^2 + \frac{1}{2} dL\eta^2\sigma^2
$$
$$
= F(w_t^{s+1}) + (c_{t+1} \left( 1 + 2\zeta\eta^2 L^2 + \eta\beta \right) + L^3\eta^2) \left\| \mathrm{Exp}_{\widetilde{w}^s}^{-1}(w_t^{s+1}) \right\|^2
$$
$$
+ \left( L\eta^2 - \eta + c_{t+1} \left( 2\zeta\eta^2 + \frac{\eta}{\beta} \right) \right) \left\| \mathrm{grad}\, F(w_t^{s+1}) \right\|^2 + \left( \frac{1}{2} dL\eta^2 + c_{t+1}\zeta d\eta^2 \right) \sigma^2
$$
$$
= R_t^{s+1} - \left( -L\eta^2 + \eta - c_{t+1} \left( 2\zeta\eta^2 + \frac{\eta}{\beta} \right) \right) \| \mathrm{grad}\, F(w_t^{s+1}) \|^2 + \left( \frac{1}{2} dL\eta^2 + c_{t+1}\zeta d\eta^2 \right) \sigma^2.
$$

Rearranging, we get

$$
\left( \eta - L\eta^2 - c_{t+1} \left( 2\zeta\eta^2 + \frac{\eta}{\beta} \right) \right) \mathbb{E}\| \mathrm{grad}\, F(w_t^{s+1}) \|^2 \leq R_t^{s+1} - R_{t+1}^{s+1} + \left( \frac{1}{2} dL\eta^2 + c_{t+1}\zeta d\eta^2 \right) \sigma^2
$$

from which we have

$$
\mathbb{E}\| \mathrm{grad}\, F(w_t^{s+1}) \|^2 \leq \frac{R_t^{s+1} - R_{t+1}^{s+1}}{\left( \eta - L\eta^2 - c_{t+1} \left( 2\zeta\eta^2 + \frac{\eta}{\beta} \right) \right)} + \frac{\left( \frac{1}{2} L + c_{t+1}\zeta \right) d\eta^2}{\left( L\eta^2 - \eta - c_{t+1} \left( 2\zeta\eta^2 + \frac{\eta}{\beta} \right) \right)} \sigma^2.
$$

$\square$

We now give the proof of Theorem 7.

*Proof.* The proof is adapted from (Zhang et al., 2016, Theorems 2, 6 and Corollary 6). Let $\delta_n = \min_t \delta_t$ and $T = mS$

$$\sum_{t=0}^{m-1} \mathbb{E} \left\| \mathrm{grad}\, f(w_t^{s+1}) \right\|^2$$

$$\leq \sum_{t=0}^{m-1} \frac{R_t^{s+1} - R_{t+1}^{s+1}}{\delta_t} + \frac{\left(\frac{1}{2}L + c_{t+1}\zeta\right) d\eta^2}{\delta_t} \sigma^2$$

$$\overset{(*)}{\leq} \frac{R_0^{s+1} - R_m^{s+1}}{\delta_n} + \frac{\left(\frac{1}{2}L + c_{t+1}\zeta\right) m d\eta^2}{\delta_n} \sigma^2$$

$$= \frac{\mathbb{E}\left[F(w_0^{s+1}) - F(w_m^{s+1}) + c_0 \left\| \mathrm{Exp}_{w_{\widetilde{s}}}(w_0^{s+1}) \right\|^2 - c_m \left\| \mathrm{Exp}_{w_{\widetilde{s}}}(w_m^{s+1}) \right\|^2\right]}{\delta_n} + \frac{\left(\frac{1}{2}L + c_0\zeta\right) m d\eta^2}{\delta_n} \sigma^2$$

$$\overset{(**)}{\leq} \frac{\mathbb{E}\left[F(\widetilde{w}^s) - F(\widetilde{w}^{s+1})\right]}{\delta_n} + \frac{\left(\frac{1}{2}L + c_0\zeta\right) m d\eta^2}{\delta_n} \sigma^2,$$

where $\delta_t \geq \delta_n$, $c_t \leq c_0$ is used in $(*)$ and that $w_0^{s+1} = \widetilde{w}^s, w_m^{s+1} = \widetilde{w}^{s+1}$ and that $c_m = 0, c_0 \geq 0$ in $(**)$.

Now, summing the gradient norm square over all the epochs and using $F(w^*) \leq F(\widetilde{w}^m)$, we get

$$\frac{1}{T} \sum_{s=0}^{S-1} \sum_{t=0}^{m-1} \mathbb{E} \left\| \mathrm{grad}\, f(w_t^{s+1}) \right\|^2 \leq \frac{\mathbb{E}\left[F(\widetilde{w}^0) - F(w^*)\right]}{T\delta_n} + \frac{\left(\frac{1}{2}L + c_0\zeta\right) d\eta^2}{\delta_n} \sigma^2.$$

Choosing $\beta = L\zeta^{1-\alpha_2}/n^{\alpha_1/2}$ and solving recurrence relation $c_t$ using $\eta, m$ given by theorem as (Zhang et al., 2016, Theorem 2) one can get $c_0 = \frac{\mu_0 L}{n^{\alpha_1/2}\zeta}(e - 1)$. Substituting that in $\delta_n \geq \frac{\nu}{Ln^{\alpha_1}\zeta^{\alpha_2}}$ and finally using this we have

$$\frac{1}{T} \sum_{s=0}^{S-1} \sum_{t=0}^{m-1} \mathbb{E} \left\| \mathrm{grad}\, f(w_t^{s+1}) \right\|^2$$

$$\leq c\frac{\mu_0 L n^{\alpha_1}\zeta^{\alpha_2}}{\nu n S} \mathbb{E}\left[F(\widetilde{w}^0) - F(w^*)\right] + \frac{Ln^{\alpha_1}\zeta^{\alpha_2}\left(\frac{1}{2}L + \frac{\mu_0 L}{n^{\alpha_1/2}\zeta}(e-1)\zeta\right)\frac{\mu_0^2}{L^2 n^{2\alpha_1}\zeta^{2\alpha_2}}}{\nu} d\sigma^2.$$

Finally, putting the values of $\alpha_1 = 2/3, \alpha_2 = 1/2\, \mu_0 = 1/10, \nu = 1/2$, and $\sigma^2 = c_2 \frac{mS \log(1/\delta)L_0^2}{n^2\epsilon^2} = c_3 \frac{S \log(1/\delta)L_0^2}{n\epsilon^2}$ one can get that

$$\mathbb{E} \left\| \mathrm{grad}\, f(w^a) \right\|^2 \leq c_4 \left( \frac{L\zeta^{1/2}}{n^{1/3}S} \mathbb{E}\left[F(\widetilde{w}^0) - F(w^*)\right] + \left[\frac{1}{n^{2/3}\zeta^{1/2}} + \frac{1}{n\zeta^{1/2}}\right] \frac{dS \log(1/\delta)L_0^2}{n\epsilon^2} \right)$$

$$\leq c_4 \left( \frac{L\zeta^{1/2}}{n^{1/3}S} \mathbb{E}\left[F(\widetilde{w}^0) - F(w^*)\right] + \frac{dS \log(1/\delta)L_0^2}{n^{5/3}\zeta^{1/2}\epsilon^2} \right).$$

Setting $S = \sqrt{\frac{L\zeta\mathbb{E}[F(\widetilde{w}^0)-F(w^*)]}{d\log(1/\delta)}} \frac{n^{2/3}\epsilon}{L_0}$, we have

$$\mathbb{E} \left\| \mathrm{grad}\, f(w^a) \right\|^2 \leq c_4 \frac{L_0\sqrt{dL \log(1/\delta)\mathbb{E}\left[F(\widetilde{w}^0) - F(w^*)\right]}}{n\epsilon}. \tag{16}$$

The gradient complexity is given by

$$S(n + 2m) = \sqrt{\frac{L\zeta\mathbb{E}\left[F(\widetilde{w}^0) - F(w^*)\right]}{d\log(1/\delta)}} \frac{n^{2/3}\epsilon}{L_0} \left(n + \frac{n}{30}\right) = \sqrt{\frac{L\zeta\mathbb{E}\left[F(\widetilde{w}^0) - F(w^*)\right]}{d\log(1/\delta)}} \frac{n^{5/3}\epsilon}{L_0}. \tag{17}$$

This completes the proof.

$\square$

### B.4.3 Proof of Theorem 8

*Proof.* With the values given in the theorem statement, $\sigma^2 = \frac{mSK\log(1/\delta)L_0^2}{n^2\epsilon^2} = \frac{Kn\lceil 6+\frac{18}{n-3}\rceil L\tau\zeta^{1/2}\frac{\mu_0}{\nu n^{1/3}}\log(1/\delta)L_0^2}{3\mu_0 n^2\epsilon^2} = \frac{K\lceil 6+\frac{18}{n-3}\rceil L\tau\zeta^{1/2}\frac{\log(1/\delta)L_0^2}{\nu n^{1/3}}}{3n\epsilon^2}$. This implies that

$$\mathbb{E}[\|\operatorname{grad} f(w^{k+1})\|^2] \leq \frac{1}{2\tau}\mathbb{E}\left[F(\widetilde{w}^0) - F(w^*)\right] + \left[\frac{1}{n^{2/3}\zeta^{1/2}} + \frac{1}{n\zeta^{1/2}}\right]\frac{dK\lceil 6+\frac{18}{n-3}\rceil L\tau\zeta^{1/2}\frac{\log(1/\delta)L_0^2}{\nu n^{1/3}}}{3n\epsilon^2}$$

$$\leq \frac{1}{2\tau}\mathbb{E}\left[F(\widetilde{w}^0) - F(w^*)\right] + \frac{24dKL\tau\log(1/\delta)L_0^2}{3n^2\epsilon^2}.$$

Using the Riemannian PL condition, we have

$$\mathbb{E}\left[f(w^{k+1}) - f(w^*)\right] \leq \tau\mathbb{E}[\|\operatorname{grad} f(w^{k+1})\|^2] \leq \frac{1}{2}\mathbb{E}\left[F(w^k) - F(w^*)\right] + \frac{24dKL\tau^2\log(1/\delta)L_0^2}{3n^2\epsilon^2}.$$

Recursively applying the above for $k = 0$ to $K-1$, we have

$$\mathbb{E}\left[f(w^K) - f(w^*)\right] \leq \frac{1}{2^K}\mathbb{E}\left[F(w^0) - F(w^*)\right] + \frac{8dKL\tau^2\log(1/\delta)L_0^2}{n^2\epsilon^2}\sum_{i=0}^{K-1}\frac{1}{2^i}$$

$$\leq \frac{1}{2^K}\mathbb{E}\left[F(w^0) - F(w^*)\right] + \frac{8dKL\tau^2\log(1/\delta)L_0^2}{n^2\epsilon^2}\sum_{i=0}^{\infty}\frac{1}{2^i}$$

$$= \frac{1}{2^K}\mathbb{E}\left[F(w^0) - F(w^*)\right] + \frac{16dKL\tau^2\log(1/\delta)L_0^2}{n^2\epsilon^2}.$$

Putting $K = \log\left(\frac{n^2\epsilon^2\mathbb{E}\left[F(w^0)-F(w^*)\right]}{dL\tau^2\log(1/\delta)L_0^2}\right)$ there is a constant $c$ such that

$$\mathbb{E}\left[f(w^K) - f(w^*)\right] \leq c\frac{dL\tau^2\log(1/\delta)L_0^2}{n^2\epsilon^2}\log\left(\frac{n^2\epsilon^2\mathbb{E}\left[F(w^0) - F(w^*)\right]}{dL\tau^2\log(1/\delta)L_0^2}\right).$$

Ignoring the log factors,

$$\mathbb{E}\left[f(w^K) - f(w^*)\right] = \mathcal{O}\left(\frac{dL\tau^2\log(1/\delta)L_0^2}{n^2\epsilon^2}\right). \tag{18}$$

Finally, the gradient complexity is given by,

$$KS(n + 2m) = \log\left(\frac{n^2\epsilon^2\mathbb{E}\left[F(w^0) - F(w^*)\right]}{dL\tau^2\log(1/\delta)L_0^2}\right)\left(\lceil 6 + \frac{18}{n-3}\rceil L\tau\zeta^{1/2}\frac{\mu_0}{\nu n^{1/3}}\right)\left(n + \lfloor\frac{n}{3\mu}\rfloor\right)$$

$$\leq L\tau\zeta^{1/2}n^{2/3}\log\left(\frac{n^2\epsilon^2\mathbb{E}\left[F(w^0) - F(w^*)\right]}{dL\tau^2\log(1/\delta)L_0^2}\right). \tag{19}$$

$\square$

## C  Additional experiments and more experimental details for Section 6

**Details on the parameter configurations of DP-RSVRG, DP-RSGD, and DP-RGD.** For DP-RGD, we tune the clipping parameters from the set $\mathcal{C} = \{1, 0.1, 0.01\}$ and the number of epochs from $\{10, 20, 30\}$. For DP-RSGD, clipping parameter is chosen from $\mathcal{C} = \{1, 0.1, 0.01\}$ and number of epochs from $\{n, n*5, n*10, n*20, n*30\}$. For DP-RSVRG number of epochs is chosen from $\{5, 10\}$ and set the frequency as $m = n/10$ and full gradient clipping parameter $\mathcal{C}$ is tuned from $\{1, 0.1\}$ and variance reduced gradient clipping parameter $\mathcal{C}_2$ from $\{1, 0.1, 0.01\}$. For all three algorithms, we tune the learning rate from $\eta = \{5e^{-5}, 1e^{-5}5e^{-4}, 1e^{-4}, \ldots, 5e^{-1}, 1e^{-1}, 1, 2, \ldots, 5\}$. In all our experiments, we use `geomstats` (Miolane et al., 2020; 2021; Myers et al., 2022)

## C.1 Details on the the Fréchet mean of SPD matrices computation and the covariance descriptors

The Riemannian distance induced by the metric is given by $\text{dist}(\mathbf{Z}_1, \mathbf{Z}_2) = \|\text{Logm}(\mathbf{Z}_2^{-1/2}\mathbf{Z}_1\mathbf{Z}_2^{-1/2})\|_{\text{F}}$, where Logm denotes matrix logarithm. Given points $\{\mathbf{Z}_1, \ldots, \mathbf{Z}_n\} \in \text{SPD}(m)$, the Fréchet mean is defined as the solution to following optimization problem: $\min_{\mathbf{W} \in \text{SPD}(m)} \left\{ F(\mathbf{W}) = \frac{1}{n}\sum_{i=1}^{n} f(\mathbf{W}; \mathbf{Z}_i) = \frac{1}{n}\sum_{i=1}^{n} \|\text{logm}(\mathbf{W}^{-1/2}\mathbf{Z}_i\mathbf{W}^{-1/2})\|_F^2 \right\}$. Riemannian gradient of $f$ is given in terms inverse Exponential map $\text{grad}\,f(\mathbf{W}, \mathbf{X}_i) = -2\text{Exp}_{\mathbf{W}}^{-1}(\mathbf{X}_i) = -2\mathbf{W}^{1/2}\text{Logm}(\mathbf{W}^{-1/2}\mathbf{X}_i\mathbf{W}^{-1/2})\mathbf{W}^{1/2}$. We take first two classes from PATHMNIST (Kather et al., 2019) (ADI, adipose tissue; BACK, background).

**Covariance descriptors.** Let $\mathcal{I} \in \mathbb{R}^{h \times w \times 3}$ denote a RGB image with height $h$ and width $w$. Let $\phi : \mathbb{R}^{h \times w \times 3} \to \mathbb{R}^{hw \times k}$ be a feature extractor of dimension $k$, i.e. $\phi(\mathcal{I})(\mathbf{x})$ is a $k$-dimensional vector at each spatial coordinate $\mathbf{x}$ in the image's domain $S$. Given a small $\eta > 0$, the covariance descriptor $R_\eta : \mathbb{R}^{h \times w \times 3} \to \text{SPD}(k)$ associated with $\phi$ is defined as

$$R_\eta(\mathcal{I}) = \left[ \frac{1}{|\mathcal{S}|} \sum_{\mathbf{x} \in S} (\phi(\mathcal{I})(\mathbf{x}) - \mu)(\phi(\mathcal{I})(\mathbf{x}) - \mu)^T \right] + \eta.I,$$

where $\mu = |S|^{-1}\sum_{\mathbf{x} \in \mathcal{S}} \phi(\mathcal{I})(\mathbf{x})$, and $\eta.I$ ensures $R_\eta(\mathcal{I}) \in \text{SPD}(k)$. Our experiments on the private Fréchet mean computation problem (Section 6.3) use the covariance descriptors with following feature vector:

$$\phi(\mathcal{I})(\mathbf{x}) = \left[ x, y, \mathcal{I}, |\mathcal{I}_x|, |\mathcal{I}_y|, |\mathcal{I}_{xx}|, |\mathcal{I}_{yy}|, \sqrt{|\mathcal{I}_x|^2 + |\mathcal{I}_y|^2}, \arctan\left(\frac{|\mathcal{I}|_x}{|\mathcal{I}|_y}\right) \right],$$

where $\mathbf{x} = (x, y)$, intensities derivatives are denoted by $\mathcal{I}_x, \mathcal{I}_y, \mathcal{I}_{xx}, \mathcal{I}_{yy}$ and $\eta = 10^{-6}$. Let $\star$ denote convolution operation, then first and second order intensity derivatives are computed as below,

$$\mathcal{I}_x = \mathcal{I} \star \frac{1}{4}\begin{bmatrix} +1 & 0 & -1 \\ +2 & 0 & -2 \\ +6 & 0 & -12 \end{bmatrix}, \mathcal{I}_x = \mathcal{I} \star \frac{1}{4}\begin{bmatrix} +1 & 0 & -1 \\ +2 & 0 & -2 \\ +6 & 0 & -12 \end{bmatrix},$$

$$\mathcal{I}_{xx} = \mathcal{I} \star \frac{1}{32}\begin{bmatrix} +1 & 0 & -2 & 0 & 1 \\ +4 & 0 & -8 & 0 & 4 \\ +6 & 0 & -12 & 0 & 6 \\ +4 & 0 & -8 & 0 & 4 \\ +1 & 0 & -2 & 0 & 1 \end{bmatrix}, \mathcal{I}_{yy} = \mathcal{I} \star \frac{1}{32}\begin{bmatrix} +1 & +4 & +6 & +4 & +1 \\ 0 & 0 & 0 & 0 & 0 \\ -2 & -8 & -12 & -8 & -2 \\ 0 & 0 & 0 & 0 & 0 \\ +1 & +4 & +6 & +4 & +1 \end{bmatrix}.$$

For RGB images, $\phi(\mathcal{I})(\mathbf{x})$ is a 11-dimensional vector that makes $R_\eta(\mathcal{I})$ a $11 \times 11$ SPD matrix.

## C.2 Details on the private leading eigenvector computation problem

The problem of computing the leading eigenvector of sample covariance matrix is $\min_{\mathbf{w} \in \mathbb{S}^m} \left\{ F(w) = \frac{1}{n}\sum_{i=1}^{n} f(\mathbf{w}; \mathbf{z}_i) = -\frac{1}{n}\sum_{i=1}^{n} \mathbf{w}^T(\mathbf{z}_i\mathbf{z}_i^T)\mathbf{w} \right\}$. It has been shown that above problem satisfies Riemannian PL condition (Zhang et al., 2016) while the problem is nonconvex in the Euclidean setting. Riemannian gradient of $f$ is given by $\text{grad}\,f(\mathbf{w}; \mathbf{z}_i) = -2(\mathbf{I}_{d+1} - \mathbf{w}\mathbf{w}^T)\mathbf{z}_i\mathbf{z}_i^T\mathbf{w}$.

