# OpenReview forum: "Improved Differentially Private Riemannian Optimization: Fast Sampling and Variance Reduction"
_TMLR — Accepted by TMLR_

### Review · Reviewer_EhVU · 2022-11-12

**Summary Of Contributions:**

The paper studies differentially private (DP) optimization on Riemmanian manifolds. They propose a new faster method of sampling from the tangent Gaussian distribution. They analyze the privacy and utility of a Riemmanian extension of DP-SVRG. Numerical experiments show that their proposed approaches offer benefits over existing methods.

**Audience:**

Yes

**Broader Impact Concerns:**

None.

**Claims And Evidence:**

No

**Requested Changes:**

See above.

**Strengths And Weaknesses:**

Overall, I think it is a fairly solid paper. However, it is not quite yet ready at this point: there is a correctness issue or two and things that need to be clarified, plus Section 5 needs a lot of work.

**Strengths:**

-Interesting problem

-*Major speedup benefits of their sampling procedure* vs. the approach of Han et al. (2022) = main strength in my opinion

-DP-RSVRG seems to offer improved practical performance over existing methods in the experiments

-Fairly clear and well-written for the most part

-*Most* results appear to be mostly correct (although I did not check the proofs carefully)

**Weaknesses [and requested changes]**

At a high level, the main weak spot of the submission in my opinion is Section 5. The algorithm doesn't seem to offer any theoretical benefits over the existing SGD algorithm, which is also simpler. Additionally, there is a seemingly unsupported claim about improved utility bounds: the bounds in Section 5 seem to essentially match the bounds of the prior work (or even be slightly worse due to dependence on the condition number, in the strongly convex/PL case). Third, I am not sure about the correctness of the privacy claim. In more detail, the weaknesses are:

*Abstract:*

-Would be great to start out with a couple of sentences introducing and *motivating* the problem of DP riemannian optimization (at a high level) to make the paper more accessible

*Intro:*

-"key step" -> "common approach" seems more apt

-Point to/reference your "side result" of improved utility given in the main body..I think it is Claim 2 right?

*Preliminaries*: It would be helpful if you could italicize or underline the key terminology/definitions.

-exponential function definition seems incorrect: there is a $\xi$ mentioned that does not appear in the definition..should be $v$ I think?

-"totally normal" is never defined. Also, "curvature is $\mathcal{W}$" is never defined

-parentheses needed in tahnh

-inconsistency with $\nu$ and $\xi$ in definition of Han et al's Gaussian distribution

*Section 3:*

-Is Claim 2 trivial or non-trivial given Han et al.'s Claim 1? I'm confused because if Han et al proved Claim 1, and Claim 2 is a direct consequence of Claim 1, then it is surprising that Han et al did not get as tight a bound on the variance as you.

-Also, why does Claim 2 not allow you to get improved utility bounds in Section 5?

-Claim 3: $\sigma^2$ is never defined!

*Section 4:*

-Writing: the transition into "Basis construction" is not smooth.

-Why can't steps 2 and 3 of Algorithm 1 be performed in stochastic optimization settings?

-analysis of Algorithm 2: Please explain (or even state as a lemma/claim, since it is pretty important) why it is a valid sampling scheme, i.e. why the output has the desired Gaussian distribution. I did not see any discussion of this

-Any conceptual/theoretical justification for why Algorithm 2 leads to speedup vs. Algorithm 1?

-Writing: transition into Subsection 4.1 is not smooth. Complete shift in gears without any warning or explanation of where/why the reader is being led.

-Writing: transition into section 5 is not smooth. For example, you could explain that next, you will use your Gaussian mechanism as a primitive to build a new variance-reduced DP Riemmanian optimization algorithm.

 *Section 5.1*

-first paragraph: first sentences don't really capture the essence of variance-reduction (VR). Also, the claim that VR methods require less gradient calls to achieve the same convergence rates as full gradient descent does not seem correct in general (or is not clearly explained): for example, in nonconvex optimization, the optimal stochastic VR methods converge at rate $1/T^{3}$, whereas gradient descent converges at rate $1/T^2$. Please clarify.

-Would help to remind the reader of what the SVRG algorithm looks like before presenting Algorithm 3. Also would help to describe DP-R(S)SGD and compare it (conceptually) vs. Algorithm 3.

-Alg 3: no noise added in line 4! Not clear if/how you can get DP without noise there. Thus, I am *unsure about the correctness of Claim 6*.

-Equation 2: (Why) can you decompose $\sigma^2$ in this way in the presence of the PT operator?

-high-level explanation for privacy proof framework is lacking. It seems you essentially rely on privacy of the Gaussian mechanism, privacy amplification by subsampling, and strong composition, right? Please explain why then you use the less transparent moments bounding approach instead.

-Notations with $\mathcal{K}$ in Claim 4 do not seem to be defined anywhere!

-Claim 6: what choice of $\sigma^2$ do you need?

-Was there any challenge in analyzing Algorithm 3? Seems to me like it would have been trivial in light of the DP-SVRG and DP-RSGD analyses.

 *Section 5.2*

-I don't see convex loss covered; so why the parentheses around (strong)?

-Theorem 8: units/scaling are wrong in utility bound. Should have some dependence on $F(w_0) - F^*$ I think.

-Discussion: why include the strongly convex result at all if the PL bound is better and attained under weaker assumptions?

-Discussion: Is there any theoretical benefit of Alg 3 over DP-RSGD? Gradient complexity is higher you say, which is *very strange/surprising* and should be explained. In the Euclidean case, (DP) SVRG has lower gradient complexity than (DP) SGD. Also utility bound of Alg 3 seems slightly inferior to DP-RSGD bound due to dependence on the condition number (ignoring log factors)?

---

### Review · Reviewer_1rhr · 2022-11-13

**Summary Of Contributions:**

The paper proposed a new noise sampling procedure for differentially private Riemannian optimization, the proposed sampling is more efficient than existing sampling procedures. In addition, the paper extends differentially private stochastic variance reduced gradient descent (DP-SVRG) to Riemannian optimization, and provided convergence analyses and experiments.



**Audience:**

Yes

**Broader Impact Concerns:**

No concerns.

**Claims And Evidence:**

Yes

**Requested Changes:**

Maybe I missed it but could authors provide some intuition on the difference of convergence rate between Riemannian version of algorithms and the original versions?

**Strengths And Weaknesses:**

Strength:
1. Improved sampling procedure is way more faster than existing ones.
2. Both convergence analyses and experiments are provided for proposed algorithms.

Weakness
1. I am not familiar with Riemannian optimization, but I feel it would be better to provide a more intuitive explanation on comparison between convergence bounds of the original version and the Riemannian optimization version.

---

### Review · Reviewer_3N95 · 2022-12-14

**Summary Of Contributions:**

The authors introduced a new method for adding noise to a Riemannian optimization procedure, where they avoided directly sampling a non-isotropic Gaussian, but instead using parallel transporting it to a new tangent space to be used instead.

**Audience:**

Yes

**Claims And Evidence:**

No

**Requested Changes:**

# On the Clarity of Specifying a Coordinate System

I have some confusion regarding section 2-3, and I would like the authors to respond to my detailed concerns below before I can comment on the contributions.

I believe the root of all confusion began in section 2, where the authors did not specify whether they are working with an intrinsic coordinate system on the manifold $\mathcal{M}$, which induces a **compatible** coordinate (basis) for the tangent space $T_w \mathcal{M}$. This is not helped by the applications in section 4, where most of the matrix manifolds were described extrinsically using an embedding into a larger Euclidean manifold.

**Question 1** Which coordinate system is the author working with in section 3?

If the authors are indeed working with the standard approach of using a coordinate system given by the manifold to begin with, then the tangent Gaussian definition can be written exactly using the induced tangent coordinates, where $\xi \in \mathbb{R}^d$ and the density is the same as Euclidean $\mathcal{N}( \mu, \sigma^2 g^{-1} )$ where $g$ is the Riemannian metric in this coordinate.

**Question 2** Then what does it mean in Claim 1 that "$\vec{\xi}$ is in its coordinate in $\mathscr{B}$"?

If the authors meant that a change of coordinate system is suggested for the manifold, then this needs to be clearly specified.

**Question 3** Is what the author proposing sampling a standard Gaussian in a specific tangent space, and parallel transporting it to a new one?

# On the Computational Cost Comparison

I understand that the authors would like avoid computing a matrix vector product form of $g^{-1/2} \xi$ where $\xi$ is a standard normal and $g$ is the Riemannian metric. And instead the authors would like sample a standard Gaussian in the tangent space, convert it back to a coordinate system (either intrinsic or extrinsic) where parallel transport can be computed after.

My questions are multiple fold:
**Question 4** Is it clear that on the manifolds the authors suggested, that this matrix vector product $g^{-1/2} \xi$ is difficult to compute? Often the metric $g$ has a lot of structure and computing its inverse or even inverse square root entry-wise can be quite fast, in which case matrix vector product can be computed quite quickly without a memory constraint. This type of computation methods is quite well studied in second order optimization methods such as natural gradient descent and K-FAC.

**Question 5** Suppose the authors can sample a standard Gaussian in the normal coordinates (where $g_{ij}(0) = \delta_{ij}$), how do the authors intend on converting this back into a coordinate system where parallel transport is known? Or do authors require that the manifold must have at least one point where the Riemannian tangent space has a trivial structure, such as SPD manifold at $I$?

**Question 6** Now that we have a tangent vector, can the authors comment on the actual cost of computing this parallel transport? For example, for the affine-invariant metric, the parallel transport formula is
$ P^{X,Y} (\eta) = (YX^{-1})^{1/2} \eta ((YX^{-1})^{1/2})^* $,
is this really cheaper given all the matrix inversions and square roots that needs computing? I believe this is at best unclear to me at the moment.

I believe this is a more serious problem whenever parallel transport is not available in closed form, which is the case for at least two of the examples given by the authors, but also generally the case for most manifolds.

# On Alternative Methods

I also have a thought on a high level on whether this method is an overkill. Firstly, is it really necessary to compute exactly a standard normal to have the desired privacy benefits, or is it just easier to analyze? I suspect that the analysis will be harder but normals are often not necessary.

Secondly, is there no other methods to sample a standard Gaussian? A projection for example onto the tangent space of a hypersphere is probably the most direct method to sample a Gaussian, I imagine there are likely other similar approaches that doesn't need parallel transporting from a fixed point to the entire manifold. The current method feels extremely complex for a basic task.

# On Additional Precision and Clarity

While less important, I believe the paper can still improve significantly in its mathematical clarity. Here are several suggestions beyond the ones I already described above regarding specifying a coordinate.

 - When the parallel transport is introduced, the authors did not specify the curve, which is usually the geodesic between two points. However, this geodesic is not always unique, in which case the authors needs to either rule out or handle the edge case.
 - Is it necessary to define sectional curvature this way? Either the reader do not already know Gauss curvature either or they do and it is not necessary.
 - When the Riemannian gradient is introduced, the authors used the notation $\langle \nabla f(w), v \rangle_2$ for the Euclidean inner product. This is very unclear as I believe the authors are suggesting this inner product is in the embedding Euclidean space, as this cannot be true in coordinates.
 - The exponential map takes two inputs (point and tangent vector), the authors should specify which part is invertible.
 - dist is undefined
 - After "there exists $\tau > 0$" there needs to be a comma or colon separating two expressions
 - $\kappa_{min}$ is undefined. And in which sense is this a curvature constant?
 - I believe $\mathcal{R}$ is a map from $\mathcal{Z}^n$ instead of $\mathcal{Z}$
 - What do the notations $\log_{o \sim \mathcal{R}(Z)}$ and $p( \mathcal{R}(Z) = o )$ mean?
 - The definition of $\mathcal{R}_i$ is unclear
 - The first introduction of $p(v) \propto \exp( -\| \xi - \mu \|_w^2 / 2\sigma^2  )$ there is no $v$ on the right hand side


**Strengths And Weaknesses:**

Strengths

I believe the authors had a clever idea - parallel transporting instead of sampling a non-isotropic Gaussian is indeed new.

Weaknesses

1. The paper in its current state is really difficult to read. In particular, I believe much of the mathematics regarding Riemannian geometry needs to be made much more precise and clear, as I had a lot of trouble trying to parse even the basic ideas the authors are proposing.
2. The authors did not make a careful comparison on the computational cost of sampling a tangent Gaussian vs. their proposed method. In its current state, it is unclear whether or not their method is actually faster.

---

> ### Author Response · Authors · 2022-12-28
> **Response Part  I**
>
> We thank for reviewer for the detailed review.  We have now done a **major revision** of the paper to address the concerns and support our claims. Please see sections 3, 4, Section 6.1 (experiments)
>
> We believe the source of confusion arises from the following two issues.
>
> **Distribution.**  We are not using the standard Gaussian distribution but are using the tangent Gaussian distribution which is intrinsically defined on a tangent space. There seems to be a confusion that the tangent Gaussian is just Euclidean Gaussian with covariance matrix defined by the metric matrix $G$, which is not the case. It should be noted that the tangent Gaussian is what is used for deriving privacy guarantees in [Han2022] and we follow the setup of [han2022]
>
> Now indeed, at \textbf{only} specific points $\hat{w}$, Riemannian metric becomes Euclidean metric. Further, coordindates of random vector follow Gaussian distribution given by  intrinsic $d = \text{dim} \mathcal{M}$, we emphasize that the equivalence holds in the \textbf{intrinsic} dimension and not in the ambient dimension. The proposed isometric transport approach is  precisely exploiting this equivalence.
>
>
> **Coordinate system.** There seems to be a confusion between the (local) coordinate system on the manifold and the (global) coordinate system of tangent space (which is the inner product space). Whenever we mean the coordinate system, it is any orthonormal coordinate system of the tangent space $T_{w} \mathcal{M}$.
>
> Now we answer Reviewers questions below,
>
> **Q1:  Which coordinate system is the author working with in section 3? If the authors are indeed working with the standard approach of using a coordinate system given by the manifold to begin with...**
>
> **A1:** We are not using any intrinsic coordinate system on the manifold. In the entire paper, when we mention the coordinate system, it is always the coordinate system of the tangent space (which is an inner product space) and this need not be tied to the intrinsic coordinate system on the manifold.
>
> It should be noted that the tangent Gaussian distribution is defined on a tangent space and the Lebesgue measure  uses (any) basis of the tangent space that is orthonormal with respect to the Riemannian metric $\langle .,. \rangle_{w}.$
>
> **Q2: Then what does it mean in Claim 1 that "$\vec{\xi}$ is in its coordinate in ${B}$"?. If the authors meant that a change of coordinate system is suggested for the manifold, then this needs to be clearly specified.**
>
> **A2:**
>
> *(First question's answer):* $\xi$ is a vector belonging to vector space $T_{w} \mathcal{M}.$  Now, given a basis $\mathcal{B} = \{\beta_{1}, \dots, \beta_{d} \}$,  there exists a unique *coordinates* vector $\textbf{a} \in \mathbb{R}^{d}$  such that $\xi = \sum_{i=1}^{d} a_{i} \beta_{i}.$
>
> *(Second question's answer):* No, we are not changing the coordinate system of the manifold. As mentioned earlier, we work with a basis on the tangent space only.
>
> **Q3: Is what the author proposing sampling a standard Gaussian in a specific tangent space, and parallel transporting it to a new one?**
>
> **A3:** No. We are proposing to sample from the *tangent Gaussian* at the current point by sampling from *tangent Gaussian* at a given reference point and transporting using *any* linear isometry, not just parallel transport, from the reference to current point.
>
> Specifically, we choose reference points such that two things happen and both are equally important for computationally efficiency:
>
> 1. The tangent space is parameterized freely, where one doesn't have to solve any system of equations, and
> 2. The Riemannian metric becomes a scaled Euclidean norm.
>
> **Q4:  Is it clear that on the manifolds the authors suggested, that this matrix vector product $g^{-1/2} \xi$ is difficult to compute?**
>
> **A4:** We are not completely sure what $g^{-1/2} \xi$ means. Now, if $g=G$ means the metric matrix of $d \times d$, then it should be pointed that this is defined *only* once a basis $\mathscr{B}$ = { $\beta_{1}, \dots, \beta_{d}$ } is chosen for tangent space $T_{w} \mathcal{M}.$ And, identifying the basis for the tangent space is the challenging part that we tackle. For example, please see Sections 3 and 4 of our revised draft. Please also see **Q4/A4** of our response to Reviewer 35Pm where we have given two examples.
>
> **Q5: Suppose the authors can sample a standard Gaussian in the normal coordinates (where $g_{ij}(0) = \delta_{ij}$), how do the authors intend on converting this back into a coordinate system where parallel transport is known?**
>
> **A5:** We are unable to understand the concern here  exactly. It will be great if $g_{ij}(0)$ be clarified as we are unable to understand its meaning.

---

> > ### Comment · Reviewer_3N95 · 2023-01-04
> > **Response**
> >
> > Thank you for the response and happy New Year.
> >
> > I'm not quite satisfied with the explanations provided above, as I am still confused. I believe the authors can yet still improve on the clarity of the Riemannian geometry aspects of this work.
> >
> > 1. Let's start with the tangent Gaussian. Surely with any definition of a Gaussian vector in the tangent space, you must also be able to describe it using an intrinsic coordinate system. Since the authors believe I am confusing the definition, what exactly is the definition of a tangent Gaussian in terms of an intrinsic coordinate system?
> >
> > 2. You also claim to not use any intrinsic coordinate system, but you then introduce a basis for the tangent space. However, any notion of transporting tangent vectors necessarily requires the basis to be compatible to a coordinate system. I am asking what is this coordinate system that you introduce this basis for? At the end of the day, I am simply confused as to how the authors can precisely define all the objects in the tangent space that follows without specifying a coordinate system.
> >
> > 3. If you change the basis of a tangent space, this implicitly changes the coordinate system of which it is compatible with, and as a result changes the definition of the tangent space Gaussian based on the coordinate. Therefore at the end of the day, how does the definition of this Gaussian change? I really cannot judge the contribution of the results without first understanding what exactly this distribution is.
> >
> > 4. In A3, you claim that you are not sampling from a standard Gaussian, but yet you claim the Riemannian metric becomes a scaled Euclidean metric. So which is it?
> >
> > Re Q4 and Q5, $g$ is the Riemannian metric in coordinate form (I am attempting to convert your notations to a standard Riemannian notation). More precisely, for any $p \in M, g_p: T_p M \times T_p M \to \mathbb{R}$ defines the inner product $\langle u, v\rangle_g = g_{ij} u^i v^j$. If you want to do a change of coordinates such that the Riemannian metric is Euclidean at zero, i.e. $g_{ij}(0) = \delta_{ij}$, then this corresponding to a normal coordinate. See https://en.wikipedia.org/wiki/Normal_coordinates
> >
> > Hopefully this helps clarify these questions. I will respond to the next part in a separate response.

---

> > > ### Author Response · Authors · 2023-01-06
> > > **Response to the Reviewer**
> > >
> > > Thanks for the quick response. At the on onset, we would like to point out difference in views of tangent space  before giving our response.
> > >
> > > **Discussion**
> > >
> > >
> > > * **Reviewers View: Tangent space from axiomatic construction.** Tangent space is  axiomatically constructed from coordinate charts (local intrinsic coordinate system of the manifold). Importantly, construction is independent of specific coordinate chart (local intrinsic coordinate system of the manifold) employed from the Atlas [Lee1997].
> > >
> > >     (a) **Bases.** From this view, the standard basis of tangent space is local coordinate basis induced by coordinate charts (local intrinsic coordinate system of the manifold).
> > >
> > >     (b) **Inner product.** From this view, the inner product $\langle \rangle_{w}$ can be described by standard basis, which is local coordinate basis vectors.
> > >
> > >
> > > * **Our View: Tangent space as a abstract inner product space.** Our view is that once it is constructed and equipped with Riemannian metric,  it is some abstract inner product space. [Han2022] and our paper takes this view of tangent space.
> > >
> > >     (a) **Bases.** From this view, the tangent space is a inner product space and there is nothing special about it . There are infinitely many bases. There is *no* standard basis.
> > >
> > >     (b) **Inner Product.** From this view, the inner product is like positive definite  bi-linear form (https://en.wikipedia.org/wiki/Bilinear_form), can be expressed again in any choosen basis.
> > >
> > >
> > >
> > > **Q1: Let's start with the tangent Gaussian. Surely with any definition of a Gaussian vector in the tangent space, you must also be able to describe it using an intrinsic coordinate system. Since the authors believe I am confusing the definition, what exactly is the definition of a tangent Gaussian in terms of an intrinsic coordinate system?**
> > >
> > >
> > > **A1:**  An intrinsic coordinate system $w = (w^1, w^2, \ldots, w^d)$ on the manifold allows to define a basis $(\partial w^1, \partial w^2, \ldots, \partial w^d)$ for the tangent space $T_w\mathcal{M}$. It is well known that any tangent vector $v$ can be represented in terms of the basis as a linear combination $v = \sum_i v_i \partial w^i$. To generate a tangent Gaussian $v$, we generate $v_i \sim \mathcal{N}(0, \sigma^2)$. Tangent Gaussian at $w$ can be defined using any orthonormal basis of tangent space. We emphasize that under our definition of measure (Definition 1, further see Remark 1 of the submission), this could be any orthonormal basis.
> > >
> > > This is the view suggested by reviewer and we do not make use of intrinsic coordinate systems on manifold in our work. Rather we view it as a abstract tangent space.
> > >
> > > **Q2: At the end of the day, I am simply confused as to how the authors can precisely define all the objects in the tangent space that follows without specifying a coordinate system.**
> > >
> > > **A2:**
> > >
> > > Please look at the discussion in the beginning of the response to see how we view the tangent space. This view point arises by treating the tangent space as a inner product vector space. Further, in the submission, we explicitly list specified bases for all manifolds of interest for better clarity in Section 4.1.
> > >
> > >
> > > **Q3: If you change the basis of a tangent space, this implicitly changes the coordinate system of which it is compatible with, and as a result changes the definition of the tangent space Gaussian based on the coordinate. Therefore at the end of the day, how does the definition of this Gaussian change? I really cannot judge the contribution of the results without first understanding what exactly this distribution is.**
> > >
> > > **A3:** No, the definition tangent Gaussian is not tied to any basis of the tangent space because measure is invariant. Equivalently, a sample generated using basis $\mathscr{B}_1$  follows same distribution as using another basis $\mathscr{B}_2$. See Definition 1, Remark 1
> > >
> > > **Q4: In A3, you claim that you are not sampling from a standard Gaussian, but yet you claim the Riemannian metric becomes a scaled Euclidean metric. So which is it?**
> > >
> > >
> > > **A4:** If by standard Gaussian, the Reviewer means sampling from $\mathcal{N}(0, \sigma^2 I_{d})$ and transforming it into suitable shape, then  indeed we are sampling from standard Gaussian at only reference points.
> > >
> > > **Refs**
> > >
> > > [Han2022] Differentially Private Riemannian Optimization
> > > [Lee1997] Riemannian geometry introduction to curvature

---

> ### Author Response · Authors · 2022-12-28
> **Response Part II**
>
>
> **Q6: Or do authors require that the manifold must have at least one point where the Riemannian tangent space has a trivial structure, such as SPD manifold at $\textbf{I}$.**
>
> **A6:** Correctness of the Algorithm 2 itself doesn't require *any* assumption. In practice, however, the computational cost indeed depends on the chosen point (reference point) and how efficiently the tangent Gaussian sample is constructed there. In Section 4 of the revised version, we have explicitly discussed various manifolds (Grassmann, Stiefel, sphere, hyperbolic, and SPD) and the chosen reference points.
>
>
> **Q7: Now that we have a tangent vector, can the authors comment on the actual cost of computing this parallel transport? For example, for the affine-invariant metric, the parallel transport formula ..., is this really cheaper given all the matrix inversions and square roots that needs computing?**
>
> **A7:**  It is $\mathcal{O}(m^3)$ for the SPD manifold with the affine-invariant (AI) metric. Furthermore, we have listed computational costs in Table 1 and also listed detailed algorithms in Section 4. See Appendix A, for the details about the isometric transportation.
>
> **Q8: I also have a thought on a high level on whether this method is an overkill. Firstly, is it really necessary to compute exactly a standard normal to have the desired privacy benefits, or is it just easier to analyze? I suspect that the analysis will be harder but normals are often not necessary.**
>
> **A8:** The tangent Gaussian distribution is used in [Han2022] for proposing DP Riemannian optimization. This form of distribution is indeed needed for deriving the privacy guarantees. See  Section 3 of[Han2022] and the discussion therein.
>
> **Q9: is there no other methods to sample a standard Gaussian? A projection for example onto the tangent space of a hypersphere is probably the most direct method to sample a Gaussian, I imagine there are likely other similar approaches that doesn't need parallel transporting from a fixed point to the entire manifold. The current method feels extremely complex for a basic task.**
>
> **A9:**This is not a standard Gaussian, but it is a tangent Gaussian. We are not sure if we can actually rigorously show that the above suggestion (to use projection) is equivalent to a tangent Gaussian and that it follows a distribution with the *density* given in [Han2022] for *general* Riemannian manifolds, (which is the setup of [han2022])
>
> Additionally, we have now given the computational costs of our proposed sampling strategy in Table 1 for a number of manifolds.  We emphasize that these are as computationally *efficient* as using projection operations.
>
> **Q10: I believe this is a more serious problem whenever parallel transport is not available in closed form, which is the case for at least two of the examples given by the authors, but also generally the case for most manifolds**
>
> **A10:** It should be noted that Claim 2 is for *any* linear isometry and not just the parallel transport. The vector transport is another kind of linear isometry which is more general than the parallel transport. It is available for every manifold by the procedure called *transportation using parallelization* [Huang2015]. For the Stiefel and Grassmann manifolds, we have in fact used vector transport given in [Huang2017] in our experiments.
>
>
> **Q11: Additional clarifications**
>
> **A11:** In the revised version, we have clarified all the comments.
>
>
> *Reference*
> [Han2022]: Differentially private Riemannian optimization.
>
> [Huang2015]: A Broyden class of quasi-Newton methods for Riemannian optimization.
>
> [Huang2017]: Intrinsic Representation of Tangent Vectors and Vector Transports on Matrix Manifolds

---

> > ### Comment · Reviewer_3N95 · 2023-01-04
> > **Response**
> >
> > I am more satisfied with the response in this section on the computational aspects. Thanks.

---

### Review · Reviewer_35Pm · 2022-12-14

**Summary Of Contributions:**

Two main contributions are claimed. First, the authors propose two algorithms for sampling from a Gaussian on the tangent space of a Riemannian manifold: (1) construct an orthonormal basis at the tangent space and construct a random linear combination of this basis where the coefficients are drawn from a Euclidean Gaussian vector; (2) construct the Gaussian at the tangent space of a fixed reference point and isometrically transport this random variable to the desired tangent space. Second, the authors provide a differential privacy analysis of a variance reduced Riemannian optimization algorithm.

**Audience:**

Yes

**Broader Impact Concerns:**

None.

**Claims And Evidence:**

No

**Requested Changes:**

In pg. 2, in the paragraph on Riemannian geometry: the description/notation of parallel transport is not precise; parallel transport depends on the choice of curve joining $w_1$ to $w_2$, so this should be specified.

Claims 1-2 are trivial; I do not understand why there is so much discussion devoted to this. Also, in Claim 2, this is not an inequality; this is an equality. I don’t think the authors should claim this is “much stronger” than the results of Han et al. (2022) when there is hardly anything that is proved (it seems that the purpose of this section is to point out a mistake of Han et al. (2022) but that can be a short note in the appendix rather than an entire section of the paper).

It is an overstatement to say that the sampling scheme introduced in this paper avoids “computationally expensive MCMC” – no one would suggest using MCMC to sample from a simple Gaussian distribution, which is what this paper tries to do. Please weaken the claims.

I have serious doubts that the implicit basis construction is useful. The authors claim that finding an orthonormal basis of the tangent space is expensive, but this is only because they use naive Gram–Schmidt orthonormalization. Note that in order to carry out the implicit basis scheme, one needs an explicit description of parallel transport, which is difficult in general. In the cases when parallel transport is explicit, it is also easy to find an explicit orthonormal basis for the metric; it seems that the authors simply have not tried. For example, for the affine-invariant metric, at the point X one can check that $\sqrt{\lambda_i \lambda_j/2} \\, (u_i u_j^\top + u_j u_i^\top)$ is an orthonormal basis for the tangent space, where $X = \sum_{i=1}^d \lambda_i u_i u_i^\top$ is the eigendecomposition; note that computing this is no harder than the matrix operations needed to compute the parallel transport. Similarly it seems to me that orthonormal bases can be constructed for the other manifolds listed in this paper, and the authors must do so in order to have a fair comparison of their method.

The word “hypersphere” should be replaced with the more customary term “sphere”.

There are many typos, of which I list just a few. In general, this submission would benefit from a thorough proofreading of the grammar. (In particular, I’d like to see typos corrected besides the ones I list below.)
- Pg. 2, in paragraph about DP-SVRG: “expanding suite…” should say “expanding the suite…”
- Top of pg. 3: “gradient of real valued” -> “gradient of a real-valued”
- Pg. 3: The second sentence of the paragraph on function classes is a run-on sentence (and in fact many of the sentences in the preliminaries section are run-on)
- Pg. 3, definition of geodesic strong convexity: should say “if for all w, w’...”
- Pg. 3, definition of curvature constant: perhaps you mean to take the absolute value of $\kappa_{\min}$?
- Pg. 3, differential privacy: the definition of the moment of the mechanism is missing an expectation symbol
- Bottom of pg. 3, “The Riemannian gradient however grad F(w)” is improper English
- Bottom of pg. 3, the density $p(\nu)$ is written with $\mu$ instead of $\nu$
- Defn. 1, the pushforward measure should be $\phi_*^{\mathscr B} \lambda$ (the $\lambda$ is missing)
- Rmk. 1, “basis” -> “bases”
- Why does the notation change from $p_w(\xi)$ to $p_\xi(\nu)$ from Defn. 2 to Claim 1?
- Claim 3 uses $\sigma_1^2$ which should be $\sigma^2$
- Pg. 8, clip operation should be defined with min not max
- Section B.1, Sampling on hyperbolic spaces: Sentence beginning with “tangent space” is not capitalized
- Pg. 21, “it’s” -> “its”

**Strengths And Weaknesses:**

I am not an expert in differential privacy, so I did not check the details of that section of the paper, but at a glance it seems to be a noteworthy contribution (although it seems not too different from the prior work of Han et al. (2022)).

However, there are several weaknesses: (1) I have doubts that the sampling part of this paper is interesting/useful, and many of the claims are overstated; and (2) the paper is very poorly written. Although the DP part of the paper seems to have enough content to merit publication, I cannot recommend acceptance until these issues are fixed.

---

> ### Author Response · Authors · 2022-12-28
> **Response Part I**
>
> We thank you for a detailed and helpful review.  We have now done a **major revision** of the paper to address all the concerns and support our claims.
>
> **1) I have doubts that the sampling part of this paper is interesting/useful.**
>
> We have revised Sections 3 and 4 thoroughly now and have detailed our sampling procedures for every manifold. We emphasize that not only the proposed transportation approach is interesting, it is also practically efficient. Furthermore, the proposed method is $2-4$ orders of magnitude faster that explicit basis baseline procedures (even if the baselines don't employ Gram-Schmidt!). See **Q4/A4.** where we expand in detail.
>
> **2) Additional experiments:**
>
> We have significantly expanded the experiments section as follows:
>
> * Sampling using the explicit basis construction approach without Gram-Schmidt for all manifolds as was requested.
> * We have added another strong baseline to further justify our claims, which is sampling using explicit basis construction with sparse operations.
>
> **3) the paper is very poorly written.**
>
> We have done major revision in this regard.
>
>
> **Q1: The description/notation of parallel transport is not precise;  parallel transport depends on the choice of curve joining to $w_1$ to $w_2$, so this should be specified.**
>
> **A1:** We have now mentioned that the parallel transport is through the geodesic curve and we use this throughout the paper.
>
> **Q2: Claims 1-2 are trivial; I do not understand why there is so much discussion devoted to this.**
>
> **A2:** We wanted to emphasize that the normalizing constant $C_{w,\sigma}$ becomes independent of the base-point once we properly define the Lebesgue measure using *orthonormal* basis, unlike it is in [Han2022]. We have revised the section to address this comment.
>
> **Q3: No one would suggest using MCMC to sample from a simple Gaussian distribution, which is what this paper tries to do. Please weaken the claims.**
>
> **A3:** We do not claim that MCMC is state-of-the-art. Furthermore, in the revised version, we have removed that part where it is written that MCMC is a standard.
>
> Our motivation for the work is to be able to move beyond both the MCMC and explicit basis construction approaches, both of which have been suggested in [Han2022]. In particular, when the Riemannian metric is Euclidean (e.g., sphere), [Han2022, Section 5] suggested the explicit basis construction approach. But when the Riemannian metric is not Euclidean, e.g., SPD with the Affine-Invariant metric, [Han2022, Section 5] suggested MCMC.
>
>
> **Q4: I have serious doubts that the implicit basis construction is useful....**
>
> **A4: This seems to be a misunderstanding that explicit is slower than transportation because of Gram-Schmidt. We emphasize that slowness has nothing to do how the bases are constructed rather it is because we need to fully enumerate the bases.
>
> In the revised version, our experiments show that the proposed sampling strategy is faster than explicit strategy even if there is no Gram-Schmidt involved and also even if we use sparsity in explicit basis construction.  In Section 6, we have the baselines: *Sampling using Gram-Schmidt*, *Sampling using explicit basis*, and *Sampling using explicit basis by exploiting sparsity*, which is a newly added baseline.
>
> Our results show that our proposed method *outperforms* the baselines by orders of magnitude.Now we justify why isometric transportation is interesting/useful. We illustrate this by two examples:
>
>  1. SPD matrices with the Affine-Invariant metric (picking the example that was pointed out) and
>  2. Grassmann manifold, where usefulness of transportation  becomes even more apparent.
>
>
> **Example 1: SPD Manifold with Affine-Invariant metric**
>
> Consider the SPD matrices of size $m \times m$ with affine invariant metric and as was correctly pointed out by the reviewer, we can indeed construct basis without Gram-Schmidt as
> $$\mathscr{B}_{ij}= \sqrt{\frac{\lambda_i \lambda_j}{2}} (\mathbf{u_i} \mathbf{u_j^{T}}+ \mathbf{u_j} \mathbf{u_i^{T}}).$$ The explicit construction of the basis would incur a computational cost of $\mathcal{O}(m^{4})$ (because there are $\mathcal{O}(m^2)$ outer products and each outer product costs $\mathcal{O}(m^2)$). On the other hand, our approach based on parallel transportation costs $\mathcal{O}(m^3)$ (see Algorithm 3 of the revised version), which is clearly faster.
>
> Another way to sample without the full construction of the basis $\mathcal{B}$, i.e, in a {implicit way}, is as $\mathbf{U} \mathbf{S} \mathbf{U}^{T}$ where $\mathbf{S_{ij}} = a_{ij} \sqrt{\frac{\lambda_i \lambda_j}{2}}$. where $a_{ij} \sim \mathcal{N}(0, \sigma^2)$ which would again cost $\mathcal{O}(m^3)$. Hence, the implicit approach is always better.

---

> ### Author Response · Authors · 2022-12-28
> **Response Part II**
>
> **Example 2 : Grassmann manifold**
>
> In this example, the benefit of isometric transportation becomes even more clearer. Consider Grassmann manifold  $\text{GR}(m,r)$ of size $m \times r $. Orthonormal basis for $T_{\mathbf{W}} \textup{GR}(m,r)$ is given by
> $$\mathscr{B_{ij}} = \mathbf{W_{\perp}} \tilde{\mathbf{e_{i}}} \mathbf{e_{j}^{T}}$$
>
> where $\tilde{\mathbf{e_{i}}} \in \mathbb{R}^{m-r}, \mathbf{e_i} \in \mathbb{R}^{r}$ are standard basis vectors and $\textbf{W}_{\perp} \in \mathbb{R}^{(m-r) \times r}$ denotes a matrix such that the columns form an orthonormal basis of the orthogonal complement of the
> column space of $\textbf{W}$
>
> * **Explicit**: computational cost of construction $\mathbf{W_{\perp}}$ whose cost is $\mathcal{O}(m(m-r)^2)$ and each (sparse) multiplication of $\mathbf{W_{\perp}} \tilde{\mathbf{e_{i}}} \mathbf{e_{j}^{T}} $ takes $\mathcal{O}(1)$ and there are $\mathcal{O}((m-r)r)$ such multiplications and finally, linear combination takes  $\mathcal{O}((m-r)r)$, the total computational cost is $\mathcal{O}(m(m-r)^2)$.
>
> * **Implicit 1**: First approach to do sampling using implcit basis construction is as following,
>   $\mathbf{U}=  \mathbf{W_{\perp}} \mathbf{B},$ where  $\mathbf{B} \in \mathbb{R}^{(m-r) \times r}$ filled with random coordinates. But again complexity of this approach is impacted by $ \mathbf{W_{\perp}}$ construction and computational cost would still be $\mathcal{O}(m(m-r)^2)$.
>
> * **Implicit 2 using isometric transportation:** There is a highly efficient way (slightly non-trivial way) to compute vector transport proposed in [Huang2017] with computation cost of $\mathcal{O}(mr^2)$, i.e., there is no dependence on $m-r$.
>
> From the above examples, we wish to convey the following points.
>
> * Sampling should be carefully done and one should avoid explicitly enumerating the bases.
>
> * The isometric transportation strategy provides a unified framework to perform sampling while also being efficient.
>
> We have significantly revamped Sections 3 and 4 in the revised version.
>
>
> **Q5: Similarly it seems to me that orthonormal bases can be constructed for the other manifolds listed in this paper, and the authors must do so in order to have a fair comparison of their method.**
>
> **A5:**  In the revised version, we have added experiments with explicit bases for all the manifolds.  We also have a new set of experiments where sampling using explicit basis is performed using sparse operations.
>
> **Q6: The word “hypersphere” should be replaced with the more customary term “sphere”.**
>
> **A6:** We have removed the term 'hypersphere' and replaced it with 'sphere'.
>
> **References**
>
> [Han2022]: Differentially private Riemannian optimization.
>
> [Huang2017]: Intrinsic representation of tangent vectors and
> vector transports on matrix manifolds.

---

> > ### Comment · Reviewer_35Pm · 2023-01-07
> > **Response**
> >
> > Thank you for your response. I am satisfied with the experiments showing that the parallel transport approach could be practically useful.

---

### Author Response · Authors · 2022-12-28
**Uploaded Revised Paper**

Dear Reviewers,


Thanks for the detailed and helfpul reviews.

We have now done a **major revision** of the paper to address all the concerns of the reviewers and to improve the overall readability of the paper. In particular, we have done the following.

   * We have revised Sections 3 and 4 thoroughly. We now explicitly mention the proposed sampling algorithms for the manifolds of interest.
   * We have added additional experiments in Section 6 (See Figure 1) as requested and in addition, a new baseline `Explicit-Sparse'.
   * We have added a discussion subsection in Section 5 comparing DP-RSVRG with DP-RSGD and DP-RGD.
   * We have proof-read the paper for typos and omissions.

Regards,
Authors

---

### Decision · Action_Editors · 2023-01-23

**Recommendation:** Accept with minor revision

**Comment:**

This is not an easy case. Two reviewers are leaning towards accept, one reviewer is leaning toward reject.
The main result (sampling from the tangent space) seems to be rather straightforward, but it improves over a previous work, so this is a simple and self-consistent result, and all reviewers like it. However,  a lot of work has been invested into the theoretical study of the DP-RSVG method, which is very technical, and it is not clear if the techniques used are novel, or are just the compilation of the previous results. Moreover, after the revision (stronger baselines for sampling), it looks like two different, unconnected contributions, and the DP part can simply be moved into another application. The title, however, suggest another focus: improving DP Riemannian optimization.
Thus, the results are good, but the title of the paper has to be fixed, as well as the necessity of the DP RO part completely is unclear.
At least, sampling has to be mentioned in the title in one way or another.
Another weak point are the numerics for the privacy guarantees. They are given on small toy examples.
To summarize, in the minor revision the following suggestions can be addressed:
1) The topic/structure of the paper regarding two different, not very well connected contribution. The strongest one is the sampling part, not the very technical DP-RO part
2) Numerical experiments on the DP-RO part look too simple.
3) In the text there are still misprints that need to be fixed.


**Audience:**

I think the there will be a certain audience, not very large, but there is an interest in Riemannian optimization and there is an interest in differential privacy.

**Claims And Evidence:**

The authors propose an algorithm for sampling from a Gaussian on the tangent space of a Riemannian manifold
by
a) building a random linear combination of the orthogonal basis vectors in the subspace
b) construct the Gaussian at the fixed space and use vector transport to the random variable on the desired tangent space.

Such Gaussian random variable is necessary for the generalization of the variance reduced optimization algorithm to the Riemannian manifolds and its differentiable privacy.
The sampling algorithm is simpler than the previous work since it does not require MCMC sampling.

The resulting algorithm allows to get improved privacy guarantee that is larger. The numerical experiment is very model and is not very illustrative.

---

> ### Author Response · Authors · 2023-02-16
> **Uploaded Camera Ready Version**
>
> We thank the AE for the comments. In the revised version, we have made the following changes as asked.
> * We have changed the title of the paper to ``Improved Differentially Private Riemannian Optimization: Fast Sampling and Variance Reduction'' to emphasize on the two contributions of the work.
>
> *  In the main text, we have mentioned that our contributions are two-fold, which may be of independent interest to the community. However, both the contributions come within the purview of differentially private Riemannian optimization framework and make it practically favorable. It should be also noted that the motivation of faster sampling comes from being able to have a better implementation of stochastic algorithms, where one needs to do repeated samplings unlike the deterministic algorithms.
>
> *  We have done additional experiments on the Fr\'echet mean computation of the Poincar\'e embeddings using DP-SVRG. We have also expanded the empirical result discussion of DP-RSVRG.
>
> * We have done proofreading of the paper.